# Paclitaxel-induced tubulin dysfunction stalls parasite development: synergistic potential with artemisinin against resistant strains

Yongxin Tang,[1,2] Xinyu Zhang,[1,2] Xiaohui He,[2] Tao Jiang,[3] Jun Cao,[1,2] Xinyu Yu[2]

ABSTRACT    In this study, paclitaxel (PTX), a well-characterized microtubule-stabilizing agent, was systematically investigated for its antimalarial potential against drug-resistant *Plasmodium falciparum*. *In vitro* experiments demonstrated that PTX potently inhibits intraerythrocytic development of drug-sensitive and -resistant strains, including chloroquine-resistant and artemisinin-resistant parasite lines, with 72-h half-maximal inhibitory concentrations ($IC_{50}$) ranging from 83.67 to 92.75 nM. PTX arrests parasite schizogony and reduces merozoite formation by ~25.69% to 37.46% when exposed to the trophozoite/schizont stage. Meanwhile, time-dependent drug activity against gametocyte maturation demonstrated that treatment during the immature gametocyte stages elicited a significant reduction in final gametocytemia (~38.24% to 58.79%), thereby validating the dual-targeting potential of PTX against both the pathogenic asexual and transmission-competent sexual parasite stages. Molecular dynamics simulations suggested that PTX was capable of binding to *P. falciparum* β-tubulin and potentially induced structural perturbations, including complex stabilization, enhanced M-loop flexibility, and global compaction, which may, in turn, disrupt microtubule dynamics. These results provide a plausible molecular basis for the antimalarial mechanism of PTX. Critically, PTX exhibits robust synergism with dihydroartemisinin (DHA) against artemisinin-resistant strains, reducing ring-stage survival to 1% and addressing the global challenge of artemisinin resistance. *In vivo* assays revealed that the PTX-DHA combination achieves near-complete parasite clearance (<1% parasitemia) and 100% survival in rodent models and suppresses parasitemia by >96% in humanized *P. falciparum* models. This finding confirms PTX's translational potential and positions it as a novel tubulin-targeting partner for DHA, offering a promising strategy to combat drug-resistant malaria while safeguarding the efficacy of artemisinin-based combination therapies.

IMPORTANCE    In the current landscape of next-generation antimalarial development, preference will be given to compounds with activity against emerging drug-resistant parasite strains. However, identifying such effective molecules through traditional diversity library screenings remains challenging. As a result, the repurposing of old drugs has emerged as a prevalent and pragmatic strategy in the search for novel antimalarial agents. Our findings highlight a significant enhancement in combinatorial therapeutic efficacy, positioning paclitaxel as a promising partner drug for overcoming emerging drug resistance in malaria parasites. Therefore, we believe that this work will confer a broad interest to the readers of different disciplines, including antimalarial development.

KEYWORDS    *Plasmodium*, paclitaxel, artemisinin resistance, synergistic effect, combination therapy

Editor Jian Li, Hubei University of Medicine, Shiyan, Hubei, China

Address correspondence to Xinyu Yu, Kimiyxy@foxmail.com, Jun Cao, caojuncn@hotmail.com, or Tao Jiang, tjiang@ibp.ac.cn.

Yongxin Tang, Xinyu Zhang, and Xiaohui He contributed equally to this article. Author order was determined according to each author's contribution to this work, including performing experiments, validating methodologies, and analyzing data.

The authors declare no conflict of interest.

See the funding table on p. 30.

Malaria, a vector-borne infectious disease, poses a significant global health burden. According to the World Health Organization (WHO) report, an estimated 241 million clinical cases occur worldwide annually, with *Plasmodium falciparum* accounting for the majority of associated deaths (1). Despite ongoing efforts to develop effective vaccines, their efficiency remains controversial (2–4), leaving drug-based therapies as the most practical and cost-effective strategy for malaria control, especially in Sub-Saharan Africa (5–7). However, the emergence and rapid spread of *P. falciparum* strains resistant to frontline antimalarial drugs have significantly hampered global malaria elimination (8–11). Resistance to multiple drug classes (e.g., chloroquine, sulfadoxine-pyrimethamine, and artemisinin) has rendered traditional treatment regimens ineffective in many endemic areas (12, 13). This alarming trend highlights an urgent need for the development of novel antimalarial compounds with a distinct mode of action (MOA) (14, 15). Such advancements are crucial to restoring therapeutic efficiency, reducing morbidity and mortality, and ultimately achieving the long-term goal of malaria elimination.

*P. falciparum* is transmitted to humans by the bite of *Anopheles* mosquitoes, in which inoculated sporozoites rapidly invade the hepatocytes (16, 17). After rounds of exponential replication, merozoites are released into the bloodstream and invade red blood cells, initiating a 48-h intraerythrocytic development cycle (IDC) (18). During IDC, parasites progress through three morphological stages: ring, trophozoite, and schizont, which requires dynamic remodeling between stable and flexible cytoskeletal states (19–21). Thus, disrupting cytoskeletal architecture represents a promising strategy to inhibit parasite proliferation, growth, and invasion. Among cytoskeletal components, microtubules, composed of tubulin protofilaments with stage-specific architectural variability in parasites, are essential for multiple cellular processes, as revealed by structural-functional analyses (22–24). Notably, microtubules have become validated targets in anticancer pharmacology, with paclitaxel (PTX) demonstrating clinical success, highlighting the therapeutic potential of targeting microtubule dynamics to disrupt the survival of proliferative parasitic cells (25–27).

As dynamic cytoskeletal structures, microtubules are essential for multiple stages of the parasite's life cycle, influencing processes such as chromosome segregation, parasite motility, organelle transport, and host-cell interaction (22, 28). During the asexual IDC of *P. falciparum*, which constitutes the symptomatic phase of human malaria, parasites undergo rapid nuclear segregation and merozoite production. Therefore, in late-stage parasites, the proper assembly and functional integrity of the mitotic spindle are critically dependent on tubulin polymers (29). Studies have shown that disrupting tubulin polymerization with antimitotic drugs (e.g., certain benzimidazoles) can induce mitotic arrest in *Plasmodium* (30). In the absence of a functional mitotic spindle, the parasite cannot effectively divide into merozoites, the invasive forms that rupture host red blood cells and infect new ones. This disruption of the asexual replication cycle significantly impairs parasite growth and propagation within the human host. Additionally, tubulin plays a critical role in gametocyte development during the sexual stage, which is essential for parasite transmission (31). It has been reported that microtubules are involved in parasite polarization and organelle rearrangement during gametocyte formation (22). The organization of the cytoskeleton, including tubulin-based microtubules, is vital for the precise positioning of organelles (e.g., the nucleus and mitochondrion) in gametogenesis. Due to its indispensable functions in parasite growth and development, tubulin represents an attractive target for antimalarial drug development. Compounds that disrupt microtubule dynamics by interfering with tubulin polymerization or depolymerization hold significant potential for the development of multistage antimalarial therapeutics.

PTX, initially isolated from the bark of the Pacific yew tree (*Taxus brevifolia*), has emerged as a paradigm-shifting chemotherapeutic agent since its discovery in the 1960s (32). Its unique mode of action, broad-spectrum anticancer activity, and subsequent development into various formulations have revolutionized the treatment of multiple malignancies (33). Studies report that PTX exerts its anticancer effects by

stoichiometrically binding to polymerized tubulin and inhibiting its depolymerization (34). This perturbation disrupts the dynamic equilibrium of microtubules, triggering cell cycle arrest as cells fail to progress through mitosis. Specifically, during metaphase, cells depend on microtubule assembly and disassembly for chromosome segregation. By locking microtubules in their polymerized state, PTX prevents functional mitotic spindle formation, halting cell division and inducing apoptosis in rapidly proliferating tumor cells (35). Given that tubulin and microtubules are essential for *Plasmodium* parasite survival, proliferation, and pathogenesis, they represent potential therapeutic targets for novel antimalarials.

Collectively, we demonstrate that the tubulin-targeting compound PTX exhibits antimalarial activity against multiple life-cycle stages of the malaria parasite, including both asexual and sexual stages during intraerythrocytic development. *In vitro* antimalarial activity of PTX against *P. falciparum* strains 3D7 (drug sensitive), Dd2 (chloroquine resistant), and 803 (artemisinin resistant) was determined using phenotype assays (growth inhibition, ring survival, and recrudescence assays). Notably, PTX displayed comparable activity against all three strains, with 72-h $IC_{50}$ values of 83.67 nM, 87.65 nM, and 92.75 nM, respectively, confirming its efficiency against drug-resistant parasites. Subsequently, molecular dynamics (MD) simulations suggested that PTX was capable of binding to the β-tubulin of *Plasmodium*, potentially stabilizing microtubule polymers and disrupting the dynamic equilibrium essential for mitotic spindle formation, which aligned with its known anti-cancer activity. Considering the synergistic potential with DHA, we quantified the combination effect both *in vitro* and *in vivo*. The PTX-DHA combination significantly reduced parasitemia to <1% in *P. berghei*-infected mice and improved survival rates to 100%, outperforming monotherapy regimens. In humanized models, the combination suppressed *P. falciparum* parasitemia by 98.7% and extended median survival days compared to monotherapy. Our study highlights the critical role of microtubule dynamicity across *Plasmodium* blood stages. Compounds that impair the equilibrium between free tubulin and polymerized microtubules can disrupt critical cellular processes such as merozoite formation, organelle transport, and gametocyte polarization, thereby inducing parasite death. Therefore, our results position PTX as a promising partner drug in combination therapies to overcome emerging artemisinin resistance.

## MATERIALS AND METHODS

### Parasite culture and asexual stage growth inhibition assay

*P. falciparum* parasite strains including wild type (3D7), artemisinin-resistant strain (803, K13^C580Y, K13^G538V), chloroquine-resistant strain (Dd2), and NF54 strain were cultured with $O^+$ erythrocytes (provided by Wuxi Blood Center) at 2% hematocrit in RPMI 1640 medium supplemented with 0.5% (wt/vol) Albumax I, 50 mg/L hypoxanthine, 25 mM $NaHCO_3$, 25 mM HEPES, and 10 mg/L gentamycin. Parasite cultures were maintained at 37°C under 5% $CO_2$, 5% $O_2$, and 90% $N_2$ as previously described (36, 37). Parasites were regularly synchronized with 5% sorbitol treatment in two consecutive cycles, and purification of late-stage parasites was achieved by 40/70% Percoll density gradient centrifugation. Parasitemia was microscopically measured based on Giemsa-stained thin blood smears.

The *in vitro* growth inhibition assay was performed using a 3-day SYBR Green I assay with slight modifications (38). Briefly, parasites were twice synchronized by 5% sorbitol at 40-h intervals, then 40/70% Percoll treatment was performed, and the remaining parasites were cultured for 3 h with fresh erythrocytes to allow merozoite invasion. Cultures were then further synchronized with 5% sorbitol to eliminate any remaining late-stage parasites and to obtain highly synchronized early ring-stage parasites (~3 h post-infection [hpi]). The parasite cultures obtained were incubated with serially diluted concentrations of PTX (ranging from 10 nM to 50 µM) at 1% parasitemia and 2% hematocrit in a 96-well black plate, and triplicate wells were prepared for each concentration. Then, parasites were cultured under standard conditions for an additional

48 h or 72 h, followed by the measurement of parasite viability. As the corresponding time point, parasites were lysed by adding 100 µL lysis buffer (20 mM Tris, 5 mM EDTA, 0.008% saponin, and 0.08% Triton X-100) containing 5× SYBR Green I solution to each well. Parasite cultures were lysed and stained in the dark for an additional 2 h at RT, and parasite-related fluorescence was monitored via microplate reader with excitation and emission wavelengths at 485 nm and 530 nm, respectively. Each plate contained a vehicle-treated and a negative control (erythrocyte only at 2% hematocrit) for background subtraction, and the growth inhibition assay was performed by two independent experiments with three replicated wells each time. Growth inhibition rate was calculated using the following equation: %inhibition = 1− [Fluorescence intensity (Treatment) − Fluorescence intensity (Negative control)]/[Fluorescence intensity (Control) − Fluorescence intensity (Negative control)]. $IC_{50}$ was calculated by fitting the data with a nonlinear regression curve using GraphPad Prism. To analyze the stage-specific antimalarial activity, highly synchronized parasites at ring (0–3 hpi), trophozoite (24 ± 2 hpi), and schizont (32 ± 2 hpi) were incubated with varied concentrations of PTX for 16 h in the 96-well plate at a starting parasitemia of 1% (39). The survival rate for the parasite was determined at 72 h by the above-mentioned SYBR Green I assay. A stage-specific parasite inhibition assay was performed in two independent experiments with three technical replicates.

## Gametocyte induction and purification

In addition to asexual stage development, phenotypic impacts of PTX for sexual stage development of *P. falciparum* were further evaluated on the NF54 strain (38, 40). The NF54 parasite line applied for gametocyte induction was continuously maintained in standardized asexual culture conditions as previously described. At 3 days prior to initiating the gametocyte induction protocol (day −3), trophozoite parasites were magnetically purified and isolated using a CytoSinct separation column (GenScript Biotech), following the manufacturer's instructions. Uninfected red blood cells were then added to the isolated trophozoites, and the culture hematocrit was adjusted to a final concentration of 1.25%. After 24 h of continuous shaking incubation under standard culturing conditions (day −2), parasite cultures were subjected to controlled nutritional stress overnight by supplementation with 50% conditioned media. On day −1, the parasitemia of the resulting parasite culture was adjusted to 3%, followed by the selective removal of spontaneously generated gametocytes via a magnetic column to eliminate pre-existing gametocyte contamination. Subsequently, parasites were aseptically transferred into gametocyte-specific culture media supplemented with 50 mM N-acetylglucosamine (NAG) (41). NAG treatment was maintained for the subsequent two reinvasion cycles to ensure the complete elimination of all asexual parasites from the culture.

## Post-induction culture and isolation of gametocytes for the immature-stage assay

On day 0, ring-stage parasites, post-spontaneous gametocyte removal, were cultured at 4% parasitemia with 2.5% hematocrit and incubated overnight. On day 1 post-induction, parasites had progressed to the mid-trophozoite stage, at which point culture medium was refreshed. Following this incubation period, non-committed asexual parasites had reverted to the ring stage, whereas committed gametocytes had differentiated into young trophozoite-like forms. Gametocyte enrichment was achieved by magnetic isolation as described above. The purity and developmental stage of the isolated gametocytes were microscopically assessed to verify gametocyte stage morphology. In preparation for use in the immature stage assay, gametocytes were transferred into 24-well plates at 1% parasitemia and 2% hematocrit and cultured in medium containing 50 mM NAG on days 1–4 to eliminate residual asexual parasites and harvested at days 5 and 6.

## Post-induction culture and isolation of gametocytes for the mature-stage assay

On day 0, ring-stage parasites were resuspended in culture medium supplemented with 50 mM NAG and adjusted to a hematocrit of 2.5%. Gametocyte incubation was in a total volume of 50 mL of parasite cultures, and 35 mL of media was replaced daily over seven successive days. On day 8, gametocytes were harvested via magnetic column purification. In preparation for use in the mature-stage assay, gametocytes were transferred into 24-well plates at 1% parasitemia and 2% hematocrit and cultured in medium containing 50 mM NAG.

## Sexual stage growth inhibition assay

A stage-specific gametocytocidal assay was conducted following 72 h of continuous drug exposure on either immature (stages II/III) or mature (stages IV/V) gametocytes, with daily culture medium replacements. Gametocytemia was assessed via Giemsa-stained thin blood smears, where all morphologically identifiable gametocytes were counted across multiple microscopic fields, regardless of their developmental stage. Untreated gametocytes, maintained in the same plates under identical conditions and processed in parallel with drug-treated groups, served as controls. To distinguish whether the observed reduction in gametocytemia resulted from direct parasite killing or developmental arrest, the percentage of gametocytes at each developmental stage was quantified in both PTX-treated and untreated control cultures. This stage-specific enumeration was performed by classifying gametocytes into their respective developmental stages during microscopic examination, providing insights into the drug's mechanism of action on gametocyte maturation.

## Quantification of intraerythrocytic merozoites

To determine whether the exposure of PTX could have an effect on the parasite progression, highly synchronized parasites at corresponding stages were incubated in the presence or the absence of PTX, and blood smears were prepared for mature schizont-infected erythrocytes (~46 hpi). The number of merozoite nuclei per schizont was microscopically counted for 30 randomly selected schizont-infected erythrocytes.

## Erythrocyte invasion assays

To determine whether the exposure of PTX could affect the invasion capability for released merozoites, an erythrocyte invasion assay was performed according to a previously reported protocol (42). Briefly, human erythrocytes were pretreated with 1 mg/mL trypsin at 37°C for 1 h under gentle shaking, followed by two washes with complete medium. Then, the soybean trypsin inhibitor was added to the packed erythrocytes and incubated at RT for 10 minutes. Pretreated erythrocytes were resuspended with an equal volume of medium and stored at 4°C until use. Mature schizonts were purified by centrifugation on 40/70% Percoll gradient, and enriched schizont-infected erythrocytes were counted by hemocytometer. $8 \times 10^5$ schizonts were mixed with trypsin-treated erythrocytes at a 1:50 ratio in 200 µL complete medium. Mixtures were added into 96-well plates with duplicate wells for each treatment and incubated under standard conditions for an additional 20 h. The invasion rate was quantified by microscopic counting of newly invaded ring-stage parasites in at least 50 microscopic fields.

## Parasite growth arrest and IDC progression assay

To analyze the arrested progression of the parasite during IDC, parasites were synchronized with two rounds of 5% sorbitol prior to assay. At day 0, ring-stage parasites were set up at 5% parasitemia, and the starting culture was divided into two aliquots. Half of the starting culture (ring-stage parasites) was incubated with PTX in 24-well plates to evaluate its effect on early-stage development. After 24 h, late-stage parasites that progressed from the starting culture were similarly applied to evaluate the PTX effect

on late-stage development. Thin blood smears were made at 0, 24, and 48 h after the addition of PTX, and microscopic analysis for growth arrest was performed by counting the number of newly invaded rings, as well as remaining trophozoites and schizonts.

To investigate the blood-stage specificity of PTX, a developmental progression assay was performed (36). Briefly, highly synchronized parasites at corresponding stages (ring, trophozoite, and schizont) were incubated with 10 µM PTX for 6 h. Parasite progression and morphological changes were monitored by Giemsa-stained thin blood smears in duplicate, with sampling at 6-h intervals over a 54-h culture period.

The growth phenotype analysis was performed to quantify the parasite invasion and egress process. Highly synchronized schizonts were co-incubated with red blood cells (RBCs) at 1% parasitemia and 2% hematocrit. Following 12 h of incubation, the quantities of ring-stage trophozoites and schizonts were quantified by counting at least 5,000 cells per experimental condition in randomly selected microscopic fields of Giemsa-stained thin blood smears. The number of ruptured schizonts and the average number of new ring-stage parasites formed per ruptured schizont were calculated using the following formulas:

Ruptured schizonts = [(Parasitemia of schizont at 0 h) − (Parasitemia of schizont at 12 h)]/(Parasitemia of schizont at 0 h)

Rings per ruptured schizont = [(Parasitemia of ring at 12 h) − (Parasitemia of ring at 0 h)]/[(Parasitemia of schizont at 0 h) − (Parasitemia of schizont at 12 h)]

## Reinvasion and recovery assay

Parasites within a 3-h developmental window were obtained via 40/70% Percoll gradient centrifugation followed by 5% sorbitol synchronization. Cultures were then plated into 96-well plates at 1% parasitemia and 2% hematocrit. Parasites were incubated with 1×, 2×, and 5× the $IC_{50}$ values at corresponding stages for 12 h, with each condition tested in triplicate. At defined time points, cultures were resuspended by gentle pipetting, and thin blood smears were prepared for each well. For the parasite recovery assay, cultures were washed twice with complete medium and resuspended thoroughly. The cultures were transferred to a new plate containing PTX-free medium and fresh erythrocytes at a 1:40 dilution, maintained under standard conditions, and parasitemia was determined on day 4.

## PfTubulin structure prediction and molecular dynamics simulation

Owing to the absence of an experimentally validated *P. falciparum* tubulin (PfTubulin) structure in public databases, homology modeling was performed using the protein sequence (PF3D7_0819900) obtained from PlasmoDB (https://plasmodb.org/plasmo/) (43). The most reliable 3D model was generated by I-TASSER (http://zhanglab.dcmb.med.umich.edu/I-TASSER) (44), and structural validation was conducted by comparing it with the AlphaFold-predicted model. Protein model pretreatment involved Gasteiger charge calculation, polar hydrogen addition, and nonpolar hydrogen merging using AutoDockTools 4.2.6, converting PDB files to PDBQT format for docking. The binding pocket was predicted via the SiteMap algorithm, and molecular docking was performed using AutoDock Vina 1.1.2. The grid box was centered at (−15.05, −12.3, 2.44) Å around the predicted pocket, with dimensions set to 20 × 20 × 20 Å to ensure comprehensive sampling. Ten docking poses were generated and ranked by binding energy (kcal/mol). MD simulation was executed using Molecular Operating Environment (MOE) software. Following protonation and energy minimization, the system was parameterized with the Amber10: EHT force field and subjected to 1,000 ns of simulation to obtain stable conformations. Trajectories were analyzed and visualized using Visual Molecular Dynamics (VMD) software, with key interactions quantified via root-mean-square deviation (RMSD) and hydrogen bond occupancy.

## *In vitro* synergistic antimalarial assay for PTX and DHA

To evaluate whether combinational use of PTX and antimalarials could effectively tackle emerging drug resistance, *in vitro* synergistic antimalarial activity was tested by incubating parasites with varied combinations corresponding to 1×, 2×, and 5× respective $IC_{50}$ values (determined in single-drug assays), with each concentration tested in 96-well plates at 1% parasitemia and 2% hematocrit (45). Drug interaction was quantified using the combination index (CI) calculated by CompuSyn software, based on the Chou-Talalay method in which CI = 1 indicated an additive effect in the absence of synergism or antagonism, CI < 1 indicated a synergism effect and CI > 1 indicated an antagonism effect (46). Experiments were performed in duplicate with three replications for each concentration.

## *In vitro* parasite reduction ratio assay

The parasite reduction ratio (PRR) assay was performed as previously described with modifications regarding the measurement of parasite viability (47, 48). Briefly, late-stage parasites were purified by CytoSinct magnetic separation column followed by 3-h normal culturing, and parasites were further synchronized with 5% sorbitol to obtain a predominant population for ring-stage parasites at 0.5% parasitemia, 1.25% hematocrit. Then parasites were incubated with DHA, PTX, CQ, and PYR at the concentration of 10× $IC_{50}$ in six-well plates, and the drug-containing medium was replenished every 24 h. Cultures at corresponding time points (0, 24, 48, 72, 96, and 120 h) were aliquoted and serially diluted into 96-well plates after washing three times with pre-warmed complete medium. Each sample was seeded with four replicates and then incubated for up to 28 days under standard conditions for parasitemia monitoring. At day 21 and day 28, parasitemia measurement was performed by HRP-2-based ELISA, and key parameters, including lag phase, log (PRR), and PCT, were calculated for *in vitro* mode of action determination.

## Ring survival assay

*In vitro* ring survival assay ($RSA_{0-3 h}$) was performed as previously described to evaluate the antimalarial activity of PTX as a partner drug against artemisinin-resistant strains (49, 50). Highly synchronized ring-stage parasites with a 3-h window (0–3 hpi) were obtained according to the above-mentioned method, and parasite suspension with 0.5%–1% parasitemia and 2% hematocrit was incubated with drugs (single DHA, single PTX, and the combination of DHA and PTX) as illustrated in the schematic of the experiment design. After culturing for the corresponding period followed by thoroughly washing with complete medium, parasites were returned to standard culture conditions for an additional 66 h in new wells. A Giemsa-stained thin blood smear was prepared and microscopically checked for parasitemia determination, in which at least 10,000 RBCs were counted. The survival rate was calculated as the ratio of drug-treated parasitemia to that of the DMSO control. Two independent experiments with three technical replicates were conducted in this assay.

To further investigate the synergistic effect of DHA and PTX against resistant strains, we then performed a modified $RSA_{0-3h}$ protocol to investigate whether the combined use could result in better performance. Briefly, 0–3 hpi parasites were exposed to single DHA or DHA-PTX combinations at concentrations corresponding to 1/4×, 1/2×, 1×, and 2× $IC_{50}$, and the survival rate was calculated as above.

For the purpose of investigating the exact time period for which was most effective against resistant parasite strains, another modified $RSA_{0-3 h}$ was performed by exposing parasites at different intraerythrocytic developmental stages to the drug combination. Similar to the stage-specific inhibition assay, synchronized parasites at the ring (0–3 hpi), trophozoite (24 ± 2 hpi), and schizont (34 ± 2 hpi) stages were first treated with a standard DHA dose for 6 h to mimic clinical exposure. Subsequently, PTX was added, and the parasites were incubated for an additional 16 h. Two independent experiments with three technical replicates were conducted in this assay.

## *In vitro* determination of parasite recrudescence

The recrudescence assay was performed following a previously reported method with minor modifications (51). Briefly, unsynchronized *P. falciparum* cultures (10 mL, 2% parasitemia, 5% hematocrit) were exposed to either 700 nM DHA or the combination of DHA and PTX for corresponding periods as illustrated in the schematic time axis. After incubation, cultures were washed twice with RPMI 1640 incomplete medium to remove the excess drug. Drug exposure was repeated three additional times at 24-h intervals, with Giemsa-stained blood smears prepared at each time point to quantify parasitemia. This generated parasite clearance and recrudescence curves. Parasitemia of control was kept under 4% by adding fresh erythrocytes. Cultures were maintained under standard culturing conditions until parasitemia reached ~4%.

## *In vivo* antimalarial activity assay in rodent malaria model

Female BALB/c mice between 6 and 8 weeks of age were purchased from Vital River Laboratory Animal Technology (Beijing, China). Mice were cultured in the experimental animal center of JIPD under specific pathogen-free conditions. Mice were housed at an ambient temperature (22°C ± 2°C) and 40%–70% humidity under 12/12-h light-dark cycle and had free access to sterile food and autoclaved water. After acclimating for 1 week, an *in vivo* parasite inhibition assay was determined against drug-sensitive *P. berghei* ANKA strain or in-house adapted resistant strain using Peter's 4-day suppressive assay (52). Detailed experimental protocols for resistance induction and corresponding phenotypic assay results are provided in the supplemental material. Mice were intraperitoneally infected with parasitized erythrocytes from a donor at a density of $1 \times 10^7$ parasitized cells/mL.

### Experiment 1: PTX monotherapy in sensitive strain-infected mice

Four hours post-infection (*P. berghei* ANKA strain), mice were randomly grouped ($n = 5$ per group) and administered intravenously with 10, 20, and 30 mg/kg PTX. The treatment lasted for three consecutive days. The negative control group received an equal volume of 0.9% saline injection.

### Experiment 2: DHA-PTX combination treatment in sensitive strain-infected mice

Four hours post-infection (*P. berghei* ANKA strain), mice were randomly grouped ($n = 5$ per group) and administered intravenously with a single treatment of 5 mg/kg DHA, 20 mg/kg PTX, or their combination. The treatment lasted for three consecutive days. The negative control group received an equal volume of 0.9% saline injection.

### Experiment 3: PTX monotherapy in resistant strain-infected mice

Four hours post-infection (in-house adapted *P. berghei*-resistant strain), mice were randomly grouped ($n = 5$ per group) and administered intravenously with 10, 20, and 30 mg/kg PTX. The treatment lasted for three consecutive days. The negative control group received an equal volume of 0.9% saline injection.

### Experiment 4: DHA-PTX combination treatment in resistant strain-infected mice

Four hours post-infection (in-house adapted *P. berghei*-resistant strain), mice were randomly grouped ($n = 5$ per group) and administered intravenously with a single treatment of 5 mg/kg DHA, 20 mg/kg PTX, or their combination. The treatment lasted for three consecutive days. The negative control group received an equal volume of 0.9% saline injection.

Blood was collected daily from the tail vein, followed by thin blood smear preparation. Giemsa-stained thin blood smears were microscopically checked to determine the parasitemia of infected mice, in which at least 5,000 cells were independently counted

by two skilled technicians. Survivability of animals was monitored up to 21 days, and the survival curve was plotted using GraphPad Prism.

### *In vivo* antimalarial assay in humanized mouse model

Female NOD/ShiLtJGpt-$Prkdc^{em26Cd52}Il2^{rgem26Cd22}kit^{em1Cin(V831M)}$/Gpt mice (herein abbreviated as NCG-X mice) between 8 and 10 weeks of age were purchased from GemPharmatech (Nanjing, China). All mice were maintained in the experimental animal center of JIPD with free access to sterile food and water under specific pathogen-free conditions. Mice were housed at an ambient temperature (22°C ± 2°C) and 40%–70% humidity under 12/12-h light-dark cycle and acclimated for 1 week before the initiation of experimental treatment.

The humanized mouse model engrafted with human red blood cells (hRBCs) was established following a published protocol (53, 54). Briefly, mice were intraperitoneally injected with 1 mL pre-warmed hRBC suspension (50% hematocrit in RPMI medium) for hRBC engraftment. hRBC reconstitution was daily assessed by determination of the level of CD235a-positive cells in the blood samples collected from the tail vein via flow cytometry. Specifically, cells were washed and incubated with PE-Cyanine7-labeled anti-human CD235a mAb (Invitrogen) and PE-labeled anti-mouse TER-119 mAb (BioLegend) for either human or mouse erythrocyte detection. Mice with stable hRBC reconstitution (≥40% human erythrocytes, confirmed 14 days post-engraftment) were infected intravenously with $1 \times 10^7$ mixed-stage *P. falciparum* parasites (WT, K13$^{C580Y}$, and K13$^{G538V}$). Once parasitemia was detected in peripheral blood (≥0.1%), mice were randomized into treatment groups ($n = 3$ per group) and administered with vehicle control, DHA (5 mg/kg), and DHA + PTX (5 mg/kg, 20 mg/kg) to evaluate the *in vivo* antimalarial efficiency. Parasitemia was microscopically quantified by Giemsa-stained smears and independently validated by two experienced researchers, in which at least 10,000 erythrocytes were counted.

## RESULTS AND DISCUSSION

### PTX exhibits *in vitro* antimalarial activity against both drug-sensitive and -resistant asexual blood-stage parasites

To investigate the *in vitro* antimalarial activity for PTX, different *P. falciparum* strains with varied drug sensitivity (WT, drug sensitive; Dd2, chloroquine resistant; 803, artemisinin resistant) were incubated with a serial concentration gradient of PTX for defined periods. The inhibition of asexual stage parasite growth for PTX was measured via a 2/3-day assay in parasites synchronized to early rings. Parasitemia was determined after incubation, and there was a drastically reduced viability for WT around 100 nM. Typical growth inhibition curves for these strains were presented in Fig. 1A, clearly illustrating that PTX effectively suppressed the viability of drug-sensitive parasite strains, with a calculated 72-h half-maximal inhibitory concentration (IC$_{50}$) of 83.67 nM. To investigate the generality of PTX against other strains, we then measured the antimalarial activity of PTX against multiple drug-resistant strains (Fig. 1B and C). Calculated IC$_{50}$ for each parasite strain lies between 87.65 and 92.75 nM, and there was no difference between strains, suggesting PTX could actively inhibit the growth of *P. falciparum,* regardless of the drug resistance. In the current landscape of next-generation antimalarial development, priority is accorded to compounds with activity against emerging drug-resistant parasite strains (55, 56). However, identifying such effective molecules through traditional diversity library screenings remains challenging. Consequently, drug repurposing has emerged as a prominent and pragmatic strategy for discovering novel antimalarial agents (57). After primarily validating the antimalarial activity of PTX, further assays have been performed to detail the profiling of its mechanism.

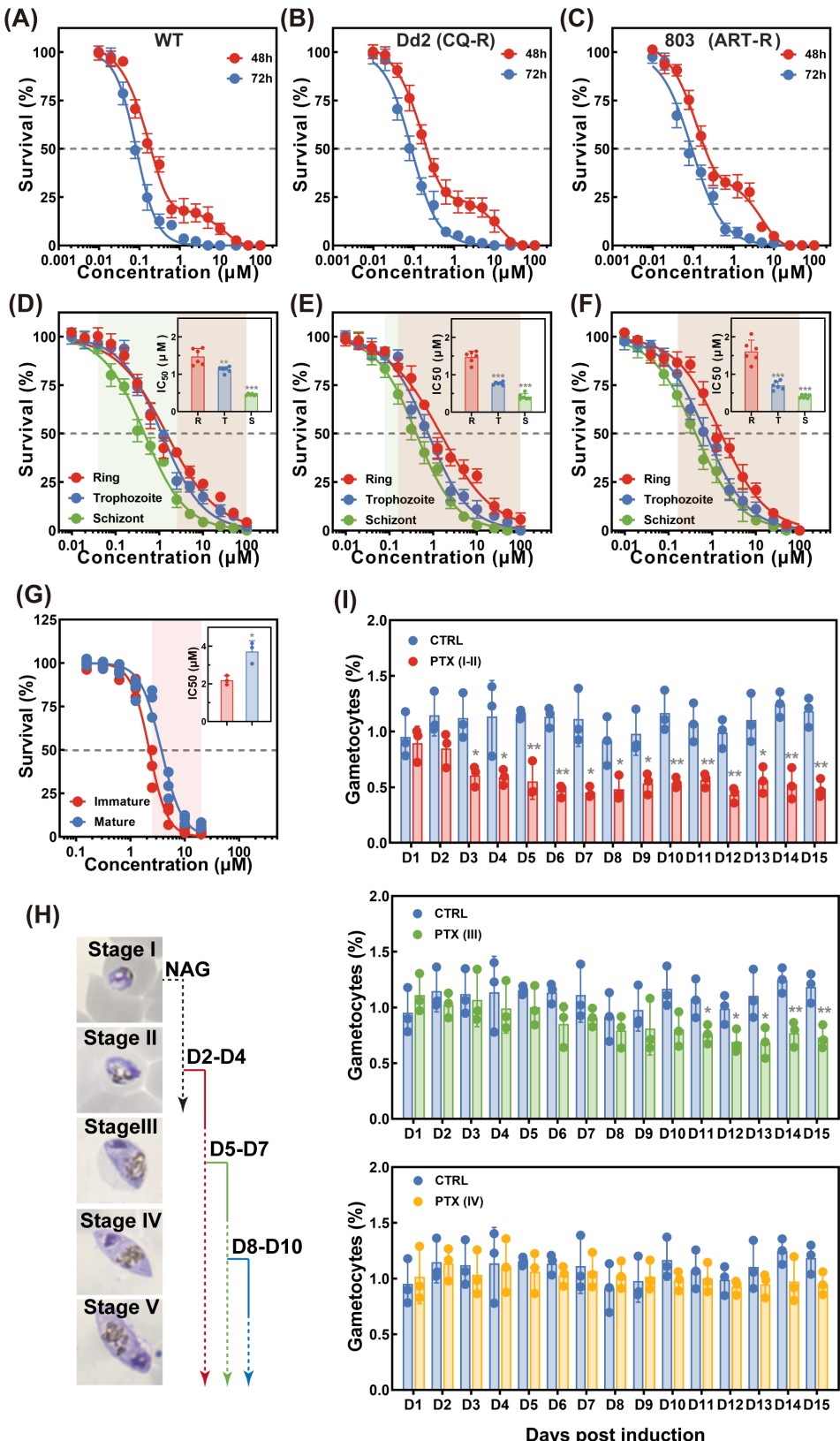

**FIG 1** Effect of PTX on *P. falciparum* during intraerythrocytic development. Typical dose-response assessments of PTX on asynchronous cultures of different *P. falciparum* strains during the asexual stage. SYBR Green I-based dose-response assay was performed to quantify parasite growth in drug-sensitive 3D7 (A), chloroquine-resistant Dd2 (B), and artemisinin-resistant 803 (C) strains. Parasites were exposed to PTX (10 nM–100 μM) for 48 h (sigmoidal dose-response fit) or 72 h ([inhibitor] vs.

**Fig 1 (Continued)**

normalized response fit) using nonlinear regression in GraphPad Prism. Typical dose-response assessments of PTX on highly synchronized cultures for evaluation of peak activity against different stages (ring, trophozoite, and schizont) of different *P. falciparum* strains, including drug-sensitive strain 3D7 (D), chloroquine-resistant strain Dd2 (E), and artemisinin-resistant 803 (F). Parasites were treated via double sorbitol synchronization and 40/70% Percoll density gradient centrifugation to isolate ring (0–3 hpi), trophozoite (24–27 hpi), and schizont (33–36 hpi) stages. The insert represented the calculated $IC_{50}$ against highly synchronized parasites at the ring, trophozoite, and schizont stages. Data were presented as mean ± SD from two independent experiments with technical triplicates. The highlighted shadow areas indicate that the parasite survival rates were statistically significant compared with those of the ring-stage group (red for trophozoite-stage parasites and green for schizont-stage parasites). Statistical significance was analyzed via one-way ANOVA followed by Bonferroni post hoc correction $*P < 0.05$, $**P < 0.01$, $***P < 0.001$. (G) *In vitro* activity of PTX against immature and mature gametocyte stages of the *P. falciparum* strain NF54 and the gametocyte viability was microscopically examined based on Giemsa-stained blood smears in which at least 5,000 erythrocytes were counted. The insert represented the calculated $IC_{50}$ for immature and mature gametocytes. Data were presented as mean ± SD from independent experiments with technical triplicates. The highlighted shadow areas indicate that the parasite survival rates were statistically significant between groups. (H) Experimental scheme to assess the impact of antimalarials on gametocytes at distinct stages of maturation. Gametocyte induction was first achieved via nutrient limitation, after which cultures were treated with NAG to selectively eliminate asexual parasite stages. Gametocytes were then magnetically enriched on day 2 post-induction to obtain stage-specific population. For drug exposure, 3-day treatments were initiated at four distinct time points post-induction: day 2 (corresponding to gametocyte stages I and II), day 5 (stage III), day 8 (stage IV), and day 11 (stage V), ensuring targeted evaluation of antimalarial effects across the full gametocyte stage. (I) Stage-specific assay for gametocytes exposed to 500 nM PTX. Data were presented as mean ± SD from independent experiments with technical triplicates. Statistical significance was analyzed using Student's t-test, with all experimental groups compared with the control group. $*P < 0.05$, $**P < 0.01$.

## Parasites display distinctive susceptibility to PTX during the IDC

During the IDC of *Plasmodium falciparum*, parasites exhibit varying degrees of susceptibility to conventional antimalarials, a phenomenon that offers valuable insights into the compound's MOA (39). Interestingly, contrasting with the monophasic curve obtained from the 72-h assay, which encompasses the entire IDC development of the parasite, a clear biphasic curve with a plateau around the range between 312.5 and 2,500 nM was observed in the 48-h growth inhibition assay. The calculated 48-h half-maximal inhibitory concentrations ($IC_{50}$) for each parasite strain were determined as 222.3, 280.2, and 287.9 nM, respectively. This unusual biphasic dose-response curve was also validated against drug-resistant strains, suggesting that PTX interacts with the parasites through a two-phase binding process (58). The plateau phase in *Plasmodium* drug growth curves is characterized by stable parasitemia with no significant fluctuation over time. It primarily originates from stage-specific differences in parasite drug susceptibility, combined with sub-inhibitory drug effects and parasite adaptive responses. Ring stages display inherently low metabolic activity, such as reduced hemoglobin digestion and ATP production. This trait minimizes the activation of artemisinin-class drugs, which rely on parasite-generated reactive oxygen species for their antiplasmodial effect, and restricts binding to the molecular targets of chloroquine-like compounds, making ring stages inherently less susceptible to drug-induced damage (39). In contrast, trophozoites and schizonts possess heightened biosynthetic activity and abundant drug targets. For example, chloroquine acts on hemozoin formation in these stages while piperaquine targets their metabolic pathways, leading to efficient inhibition by therapeutic drug concentrations (59, 60). This creates a dynamic balance between the death of trophozoites/schizonts and the survival of ring stages, which sustains the plateau phase. Compounding this issue, sub-inhibitory drug concentrations often fail to saturate all molecular targets. Chloroquine, for instance, cannot fully inhibit heme detoxification in ring stages, and piperaquine treatment may result in incomplete growth inhibition even at higher concentrations. Meanwhile, adaptive mechanisms such as upregulation of heat shock proteins further enhance ring-stage survival under drug pressure (61).

Notably, the plateau often overlaps with bimodal growth (two distinct peaks) due to asynchronous parasite development. Initial inhibition of drug-sensitive trophozoites and schizonts forms the first peak, followed by the proliferation of dormant ring stages that evade initial drug exposure and mature into replicative forms to form the second peak. The plateau observed in shorter assays disappears in 72-h drug exposure experiments, a phenomenon attributable to the cumulative pharmacodynamic effects of PTX that span the full 48-h erythrocytic life cycle of the parasite. Specifically, the initially present ring and trophozoite stages mature into schizonts over time, and these schizonts are inherently sensitive to PTX. As a result, all parasite stages eventually become susceptible to the drug, abolishing the stage-specific segregation of inhibitory activity and yielding a single, well-defined sigmoidal dose-response curve. Collectively, these interconnected factors converge to shape the plateau phase, a hallmark of the complex interaction between antimalarials and *Plasmodium*'s developmental biology and adaptive capabilities.

This could be attributed to stage-specific differential susceptibility of the parasite or polypharmacology effects resulting from varied MOAs (62). Understanding the precise MOA of antimalarial compounds is crucial, as it can provide valuable insights into their clinical efficacy in clearing parasites from the bloodstream. Given that PTX inhibits parasite development and produces biphasic dose-response curves, we hypothesized that PTX may exhibit enhanced activity against late-stage parasites. To test this hypothesis, we conducted stage-specific assays to determine the peak activity of PTX against different stages of the *P. falciparum* during IDC. Highly synchronized parasites at corresponding stages with a ~3-h window were exposed to either PTX or vehicle control at different time intervals, as illustrated in the Materials and Methods section. Then, the cultures were transferred to new plates and returned to normal conditions after washing with medium to monitor the parasite progression. The time course of parasite viability across different stages, after incubation with PTX, is illustrated in Fig. 1D through F. At 72 hpi, all PTX-treated parasites showed a significantly reduced survival rate compared to the control group. However, the most pronounced reduction in parasitemia was observed when PTX was administered during the trophozoite or schizont stages ($P < 0.001$, analyzed by one-way ANOVA with Bonferroni post hoc pairwise comparisons between each group). The exact $IC_{50}$ values are as follows: ring stage = 1.47 ± 0.19, 1.52 ± 0.14, and 1.61 ± 0.27 µM; trophozoite stage = 1.12 ± 0.07, 0.77 ± 0.04, and 0.69 ± 0.10 µM; and schizont stage = 0.46 ± 0.02, 0.41 ± 0.07, and 0.40 ± 0.04 µM. This indicates that the peak activity of PTX is specifically directed against late-stage parasites during the IDC in which tubulin played an important role during this process (22). This finding aligns well with the biphasic dose-response curve, suggesting that PTX may arrest the development of blood-stage parasites by disrupting the normal function of tubulin. In contrast, a 72-h incubation with PTX exerted a cumulative effect on parasites across all stages, resulting in a single sigmoid dose-response curve.

Considering the asexual blood stage of *Plasmodium* parasites is the primary driver of malaria pathogenesis, which is responsible for clinical symptoms and disease transmission (63); therefore, stage-specific assays during the IDC are essential for comprehensively evaluating both the efficacy and MOA of antimalarials. During IDC, different parasite stages (ring, trophozoite, and schizont) exhibit unique biological vulnerabilities. Trophozoites undergo active metabolism and hemoglobin digestion within the RBC, while schizonts, on the other hand, are engaged in the process of nuclear division to produce merozoites (20, 64, 65). By inhibiting either the transition between parasite stages or the formation of daughter merozoites, it is possible to prevent subsequent rounds of infection in the bloodstream. Stage-specific assay has not only revealed the peak activity, but also provided critical information for developing combination therapies, where drugs with different stage-specific activities can be combined to achieve a more comprehensive treatment effect.

## PTX disrupts gametocyte development and formation during the sexual stage

To comprehensively evaluate whether PTX could inhibit the progression of parasite sexual development, a crucial step for malaria transmission from human hosts to *Anopheles* mosquitoes, we conducted a gametocyte inhibition assay in the presence and absence of PTX at a $5\times$ $IC_{50}$ concentration (previously determined for asexual stage parasites). The cytoskeleton of *Plasmodium* gametocytes plays an indispensable role throughout both the immature and mature phases, with substantial differences in its functional priorities between these stages. During the immature stage, cytoskeleton-driven structural remodeling and developmental regulation predominate, whereas cytoskeletal reorganization to support transmission and gamete formation is the central focus in mature gametocytes (66, 67). Therefore, supplementary experiments were conducted by exposing gametocytes at stages I, II, III, and IV, following the methods described in a previous study (41). As shown in Fig. 1G, our findings further demonstrated that immature gametocytes were significantly more susceptible to PTX than mature gametocytes, with corresponding 72-h $IC_{50}$ values of 2.13 μM and 3.43 μM, respectively ($P < 0.05$). The maturation of *Plasmodium* gametocytes, progressing from early round forms to mature spindle-shaped or crescent-shaped gametocytes, is critically dependent on the structural remodeling and functional modulation of subpellicular microtubules (SPMTs). These structures are indispensable for gametocyte morphological specialization, maintenance of developmental homeostasis, and acquisition of mosquito infective potential. Early immature gametocytes (stages I–III) exhibit a spherical morphology, with SPMTs distributed in a diffuse and disorganized pattern beneath the plasma membrane; upon transitioning to the mature phase (stages IV and V), SPMTs undergo rapid and coordinated assembly into parallel, aligned, rigid microtubule bundles. These bundles extend along the cell's longitudinal axis and anchor to the apical complex domains at both cellular poles, directly orchestrating the morphological transition from a spherical to sex-specific specialized phenotype. Experimental evidence confirms that pharmacological inhibition of SPMT polymerization induces gametocytes to arrest in a persistent spherical morphology, prevents the completion of morphological specialization, and ultimately results in an over 80% reduction in gametocyte maturation efficiency. Therefore, we propose that PTX treatment disrupts tubulin dynamics and impairs the assembly of the subpellicular microtubule network, leading to impaired gametocyte development and ultimately exerting a lethal effect on gametocytes.

Additionally, PTX exposure during immature gametocyte stages (stages II–III) elicited a pronounced reduction in gametocytemia, whereas only minimal inhibitory effects were observed following exposure at mature gametocyte stages (Fig. 1H and I). These results are consistent with our prior stage-specific susceptibility assays, which demonstrated that parasite sensitivity to PTX varies significantly throughout gametocyte development. The cytoskeleton of *Plasmodium* gametocytes plays indispensable roles throughout both immature (stages I–III) and mature (stage IV/V) phases, yet their functional focuses differ significantly. The immature stage is dominated by cytoskeleton-driven structural shaping and developmental regulation, while the mature stage is centered on cytoskeleton reorganization to serve transmission and gamete formation. Specifically, for the immature parasites, they undergo a dramatic shape shift from spherical to elongated. The subpellicular microtubule network, together with alveolin-related intermediate filament-like proteins, forms the pellicle structure. This structure endows the immature gametocytes with rigidity and determines their basic elongated morphology. Disruption of this microtubule network will lead to abnormal gametocyte morphogenesis. While for the mature gametocytes, the cytoskeleton is crucial for gamete production after mature gametocytes enter the mosquito midgut. In mature gametocytes, via cytoskeletal remodeling and functional specialization, it converges on the terminal stages of gametogenesis and transmission. While their functions are distinct and stage-specific, both collectively underpin the complete life cycle of gametocytes, from initial differentiation to full maturation. This finding aligns with previous reports showing that

tubulin plays a key role in the structural integrity and developmental processes of early gametocytes (68).

Collectively, these findings provide compelling evidence that PTX exerts its antimalarial effects on both the asexual and sexual stages of the *Plasmodium* life cycle by interfering with tubulin-related functions. During the asexual blood stage, PTX disrupts normal cell division processes, while in the sexual stage, it impairs gametocyte maturation.

## PTX treatment impairs parasite proliferation by disrupting schizogony and merozoite formation

To characterize the detailed MOA of PTX on parasite proliferation, we performed phenotypic assays to monitor schizogony dynamics to determine whether PTX treatment impaired the parasite proliferation (Fig. 2A). Apparent differences in parasite schizogony were observed: PTX-treated parasites at trophozoite or schizont stages produced significantly fewer merozoites per schizont compared to vehicle controls (Fig. 2B through D). Specifically, mature schizonts in untreated cultures released 19 merozoites on average, whereas PTX-exposed schizonts released only 13 merozoites, a ~32% reduction in merozoite yield. The asexual replication cycle of *Plasmodium* relies on exponential expansion via merozoite release and red blood cell invasion. Antimalarials targeting merozoite biogenesis in mature schizonts can disrupt this cycle (69). This disruption not only leads to a reduction in the number of viable merozoites formed within the schizont but also results in structurally aberrant merozoites with compromised apical complexes, and this can directly impact the parasite's ability to invade new red blood cells. As a result, the subsequent invasion of new RBCs is limited, and the asexual replication cycle is interrupted.

To investigate whether the marked reduction in parasitemia resulted from decreased multiplication or invasion efficiency, we tracked growth profiles of parasite strains exposed to PTX at different IDC stages. Consistent with reduced merozoite numbers, PTX treatment at trophozoite/schizont stages caused a significant decrease in multiplication rate: vehicle-treated parasites multiplied at 4.5-fold, while PTX-exposed cultures showed only 3.4-, 2.8-, and 2.8-fold multiplication, respectively (Fig. 2E). In addition, the percentage of ruptured schizonts dropped from 89.7% to 60.3% in PTX-treated parasites (Fig. 2F), and the number of new rings per ruptured schizont decreased from 3.3 to 2.8 (Fig. 2G). Unruptured schizonts with <8 nuclei increased fourfold, further indicating PTX arrests late-stage development.

To determine whether the reduced multiplicate rate resulted from the limited invasion capacity, the invasion efficiency of parasites was further quantified. Invasion efficiency assays using heparin as a positive control revealed no significant difference between PTX-treated and vehicle-control parasites, ruling out impaired invasion as the cause of reduced proliferation (Fig. 2H through J). Instead, microscopic analysis showed PTX-induced delayed nuclear division in trophozoites and schizonts, confirming that PTX impairs parasite growth by disrupting early-to-late-stage progression rather than invasion/egress. Collectively, these findings demonstrate that PTX targets the microtubule-dependent biogenesis of merozoites in schizonts, thereby interrupting the asexual replication cycle. This mechanism not only clarifies PTX's mode of action against asexual blood stages but also underscores its potential as a dual-action agent with activity against both asexual blood stages and sexual gametocyte development.

## Growth inhibition and defective progression of *P. falciparum* upon PTX treatment

To further characterize whether PTX exposure could impair progression during IDC, the dynamic developmental profiles of parasite development had been investigated for synchronized parasite cultures at 6-h intervals (Fig. 3A). Morphological observations of PTX-treated parasites revealed a significant disruption in IDC progression. Parasites exposed to PTX at various developmental stages exhibited a pronounced arrest in the

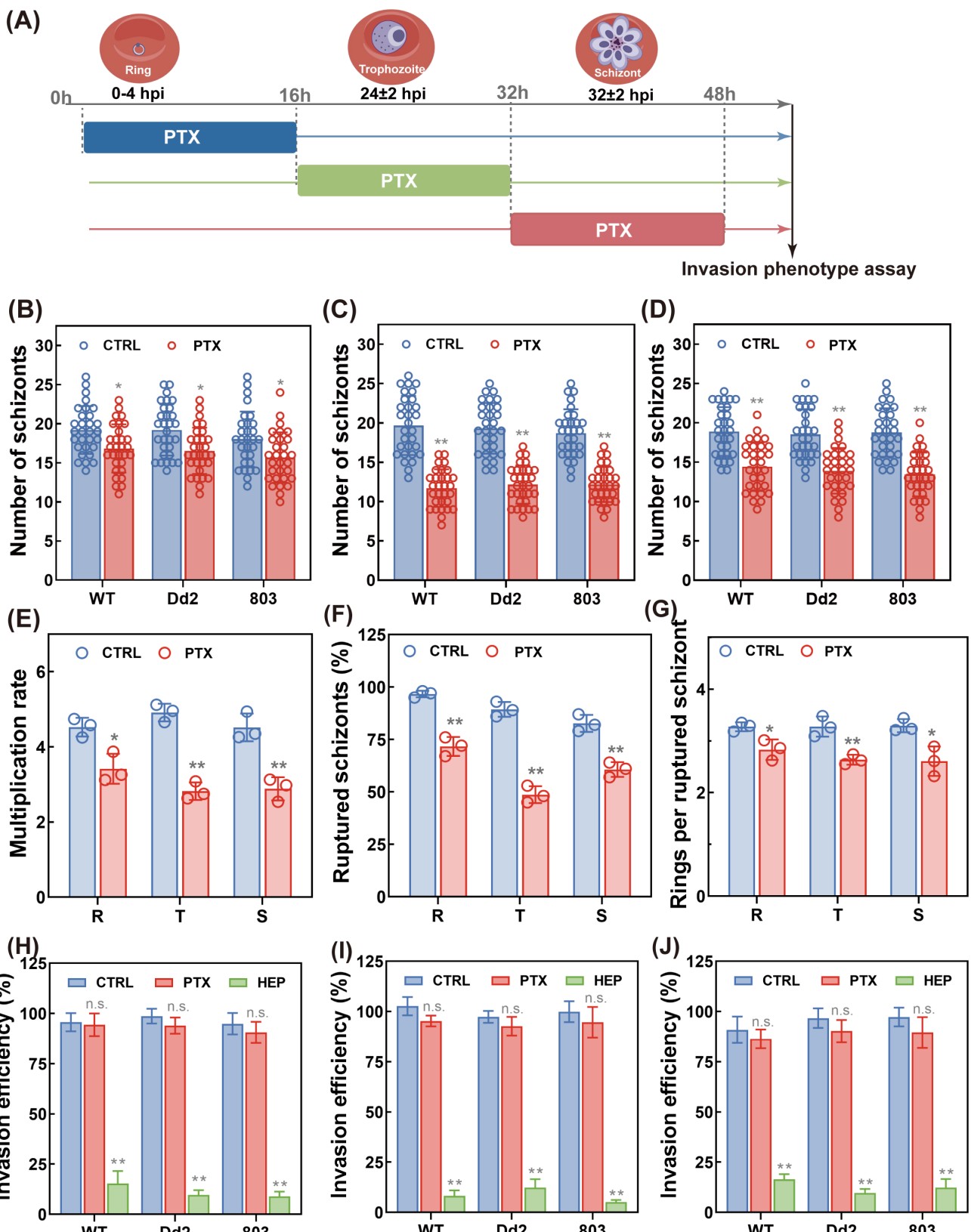

**FIG 2** Phenotypic characterization of PTX treatment on merozoite formation and re-invasion during intraerythrocytic development. (A) Schematic illustration of the corresponding stages of *P. falciparum* for invasion phenotype assay, in which the ring stage referred to parasites 0–4 hpi, the trophozoite stage referred to those at 24 ± 2 hpi, and the schizont stage referred to those at 32 ± 2 hpi. Merozoite count assay (B–D): Highly synchronized parasites at the ring (Continued on next page)

Fig 2 (Continued)

(B), trophozoite (C), and schizont (D) stages were exposed to PTX or vehicle control for 16 h. The number of merozoites per schizont was microscopically quantified from 30 mature schizonts (~45 to 48 hpi) with clearly distinguishable nuclei. Parasite multiplication rate (E): The multiplication rate of the 3D7 strain was determined following PTX exposure at the aforementioned developmental stages, defined as the fold increase in parasitemia per 48-h cycle, measured over three consecutive cycles to capture long-term growth dynamics. Schizont rupture efficiency (F): The percentage of ruptured schizonts was quantified in the 3D7 strain after PTX treatment at each target stage, reflecting the drug's impact on schizont maturation and egress. Ring formation per ruptured schizont (G): The number of newly formed rings per ruptured schizont was determined as described above, serving as a downstream readout of invasion success. Invasion efficiency assay (H–J): Comparison of PTX or vehicle treatment effects on *P. falciparum* merozoite invasion into erythrocytes, using highly synchronized parasites at the ring (H), trophozoite (I), and schizont (J) stages. Heparin, a known inhibitor of essential erythrocyte invasion events, was included as a positive control to validate assay specificity. Data were obtained from three independent assays with two technical replicates each time and presented as mean ± SD. *P*-values were determined using the two-tailed student's t-test. *$P < 0.05$, **$P < 0.01$ and n.s. indicating no significant difference.

transition from early to late stages (Fig. 3B). In control cultures, the IDC typically spanned approximately 48 h, characterized by a sequential progression from ring to trophozoites, schizonts, and finally, the release of merozoites (Fig. 3C through E). In contrast, PTX-treated cultures displayed a distinct growth arrest phenotype. Schizont-stage parasites treated with PTX showed a developmental delay, evidenced by reduced numbers of newly invaded rings in the subsequent infection cycle (Fig. 3F through H). When PTX was applied during the schizont stage, mature schizonts with abnormal nuclear segmentation were observed, and the emergence of new ring-stage parasites was severely suppressed, with minimal ring formation even at 54 hpi. Microtubules play a critical role in multiple key processes during the IDC. During the transition from trophozoites to schizonts, microtubules are essential for proper nuclear segregation, organelle positioning, and merozoite formation (22). PTX, known to promote tubulin polymerization and stabilize microtubules, disrupts these processes by inducing the formation of abnormally stable microtubule bundles (26). This interference likely impairs the dynamic reorganization of the cytoskeleton required for normal parasite development.

To monitor the early-to-late transition, the growth inhibition and defective progression of parasites at different stages upon PTX treatment were microscopically quantified. Parasites treated with 10 µM PTX exhibited a marked defect in developmental progression compared to vehicle controls (Fig. 3I through K). At 36 hpi, regardless of the initial exposure stage, PTX-treated cultures showed a significant reduction in the number of mature trophozoites and schizonts. Between 42 and 48 hpi, the relative percentage of late-stage parasites (trophozoites and schizonts) was substantially lower in PTX-treated cultures compared to controls. In the subsequent infection cycle, PTX-treated cultures at the trophozoite and schizont stage formed 85% fewer ring-stage parasites, indicating that PTX not only inhibits the development of parasites within a single IDC but also impairs the establishment of subsequent infection cycles.

## PTX delays progression of ring- or trophozoite-treated asexual blood-stage parasites

To characterize the MOA of PTX, parasite progression was specifically quantified by evaluating the effects of drug exposure on subsequent parasite development through the IDC (70). Highly synchronized ring-stage parasites (~3 hpi) were subjected to either 24 or 48 h of PTX exposure, and parasite development was monitored using Giemsa-stained smears, with 5,000 erythrocytes counted per slide to determine stage-specific parasitemia. As shown in Fig. 4A and B, vehicle control-treated rings progressed normally to trophozoites/schizonts, with parasitemia increasing from 1% to 4% over 48 h, reflecting successful merozoite reinvasion. Strikingly, PTX exposure for 48 h caused a 50% reduction in new ring formation, accompanied by a sixfold increase in unruptured schizonts with abnormal nuclear morphology. In contrast, 24-h PTX treatment of rings showed no significant effect on early-stage development but impaired subsequent schizogony, as evidenced by reduced merozoite release in the next IDC.

To further investigate whether late-stage parasites exhibit higher susceptibility to PTX, the progression of trophozoites through subsequent IDC under 24/48-h PTX

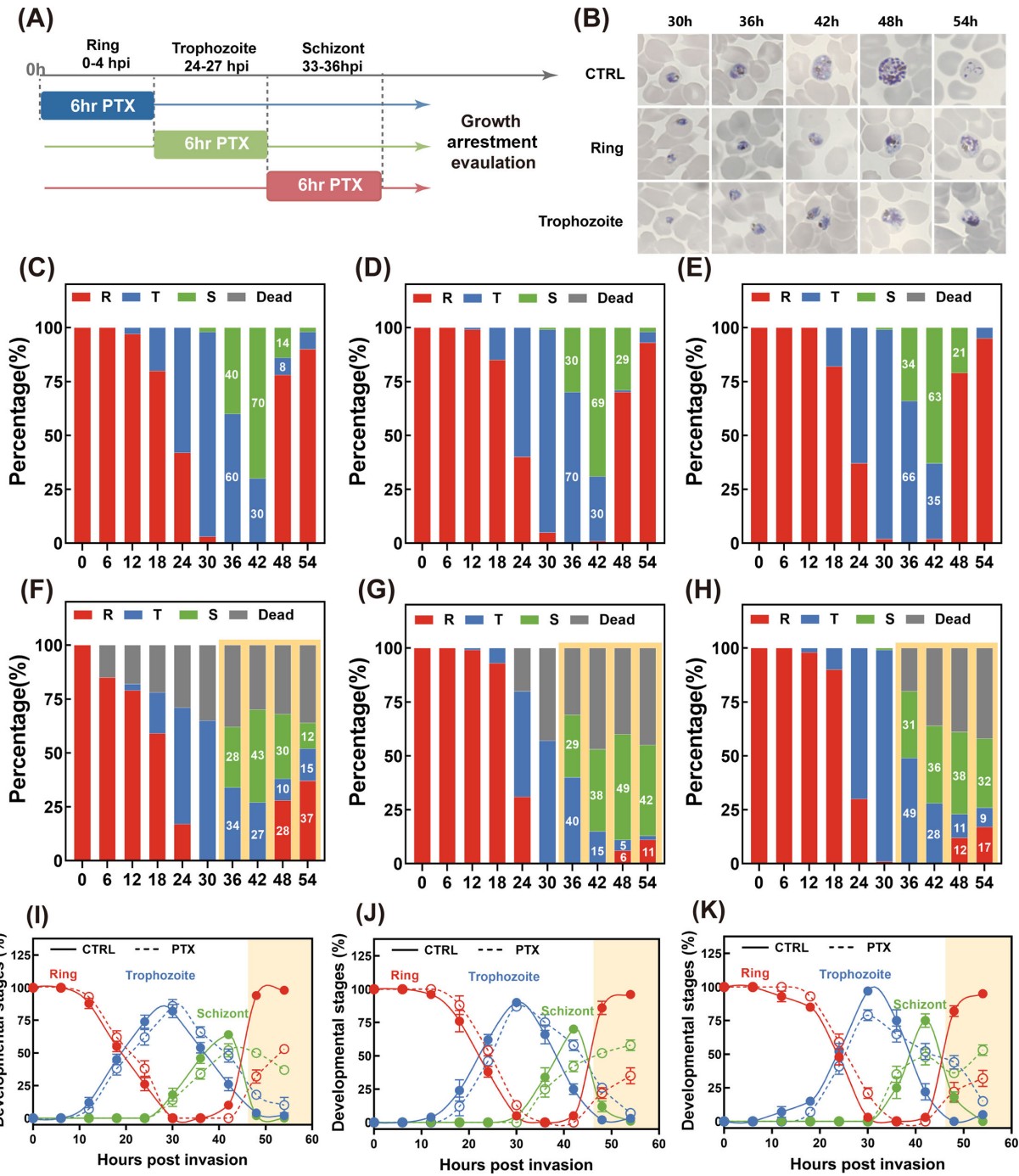

**FIG 3** Phenotypic characterization of PTX exposure across the asexual life cycle. (A) Schematic illustration of the experimental design for evaluating *P. falciparum* growth arrest following a 6-h exposure to PTX. Parasite cultures were synchronized using double 5% sorbitol treatment to isolate homogeneous populations of distinct developmental stages, enabling stage-specific drug exposure and subsequent growth assessment. (B) Representative Giemsa-stained microscopic images showing morphological characteristics of 3D7 strain parasites with arrested IDC development. Parasites were transiently exposed to PTX or vehicle control at either the ring or trophozoite stage (scale bar = 5 µm). (C–E) Quantitative analysis of morphological distribution in vehicle control-treated parasites (*n* = 100 parasites counted per group) following exposure at the ring (C), trophozoite (D), and schizont (E) stages. At each time point, parasites were morphologically categorized into four groups (ring, trophozoite, schizont, or dead form), and percentages were determined by microscopic examination to reflect normal IDC growth progression. (F–H) Quantitative morphological distribution of parasites (*n* = 100 counted per group) after 6-h PTX exposure at the ring (F), trophozoite (G), and schizont (H) stages, using the same categorization criteria as the control group. Observed growth arrest was labeled in color background, and the percentage of parasites at corresponding stage was numerically displayed in each figure. (I–K) Temporal progression of IDC stages in synchronized 3D7 parasites (Continued on next page)

Fig 3 (Continued)

treated with PTX versus vehicle control. For parasites exposed to PTX or control at the ring (I), trophozoite (J), and schizont (K) stages, the percentage of parasites in the ring (red), trophozoite (blue), and schizont (green) stages was monitored at 6-h intervals over a 54-h period. These line graphs illustrate dynamic changes in IDC progression and the extent of PTX-induced growth arrest across different developmental phases.

treatment was monitored (Fig. 4C and D). Trophozoite-treated cultures exhibited arrested schizont development at 48 h, with only 45% of parasites developing into the subsequent cycle, compared to 93% in controls. Additionally, parasitemia in PTX-treated trophozoites remained at 1.8%, whereas controls reached 4.7%, confirming that late-stage parasites are more vulnerable to PTX. Concurrently, the majority of parasites in the vehicle control group progressed to trophozoites or schizonts within 48 h, whereas only a small fraction entered schizogony in PTX-treated cultures. Collectively, these data demonstrate that prolonged PTX exposure at late IDC stages potently arrests parasite progression by disrupting microtubule-dependent cytokinesis.

## Transient PTX exposure affects the parasite IDC development with stage-specific susceptibility

To determine whether transient PTX exposure elicits persistent growth arrest during the IDC, synchronized *P. falciparum* cultures were exposed to increasing PTX concentrations for 16 h at defined post-infection time points, as outlined in the scheme (Fig. 5A). Proliferative capacity was assessed by monitoring stage transitions via Giemsa-stained thin smears. Contrasting with vehicle controls, where the majority of parasites developed into new rings with minimal residual trophozoites/schizonts by 50 h post-infection, transient exposure to 5× $IC_{50}$ PTX caused a 39.6% reduction in new ring formation (Fig. 5B). This phenotype was dose-dependent: 10× $IC_{50}$ PTX reduced rings by 40.1%, while the proportion of unruptured schizonts increased from 22.3% to 67.6%. Microscopic analysis revealed that DMSO-treated parasites completed reinvasion with normal schizogony, whereas PTX-exposed cultures accumulated trophozoites with abnormal nuclear lobing and schizonts with disrupted merozoite budding. To distinguish between developmental delay and irreversible arrest, the kinetics of developmental arrest following PTX pulsing was measured by tracking late-stage parasites after PTX treatment at 52 hpi. PTX-treated cultures showed an increase in remaining trophozoites and schizonts, suggesting that PTX treatment blocks parasite transition (Fig. 5C and D). Lower PTX doses induced delayed schizont accumulation, while higher doses significantly arrested development at the trophozoite stage, a phenotype consistent with dose-dependent microtubule stabilization.

Recovery assays were conducted to assess whether transient PTX exposure induces long-term effects on parasite proliferation and whether such effects are stage-specific. After drug removal, cultures were diluted 1:40, and parasitemia was quantified after 120 h to assess long-term effects (Fig. 5E). PTX exposure at the ring stage reduced total parasitemia by 15.5%, whereas late-stage (trophozoite/schizont) exposure caused a 54.7%–64.8% reduction compared to vehicle controls. High-dose PTX induced stage-nonspecific long-term arrest, likely due to cumulative microtubule disruption. Collectively, these findings demonstrate that transient PTX exposure elicits both acute and prolonged IDC arrest, with late-stage parasites displaying markedly enhanced sensitivity to PTX.

## MD simulation reveals effects of PTX on tubulin

To dissect the molecular basis of PTX-mediated antimalarial activity, this study integrated homology modeling and 1,000 ns molecular dynamics simulations to characterize PTX binding to PfTubulin. Given the lack of experimentally resolved PfTubulin structures, a reliable 3D model was constructed using the PF3D7_0819900 sequence (from PlasmoDB) via ITASSER, validated against AlphaFold predictions, and parameterized with the Amber10:EHT force field in MOE software, enabling quantitative analysis of structural dynamics critical to understanding PTX's impact on parasite development (Fig. 6A). MD

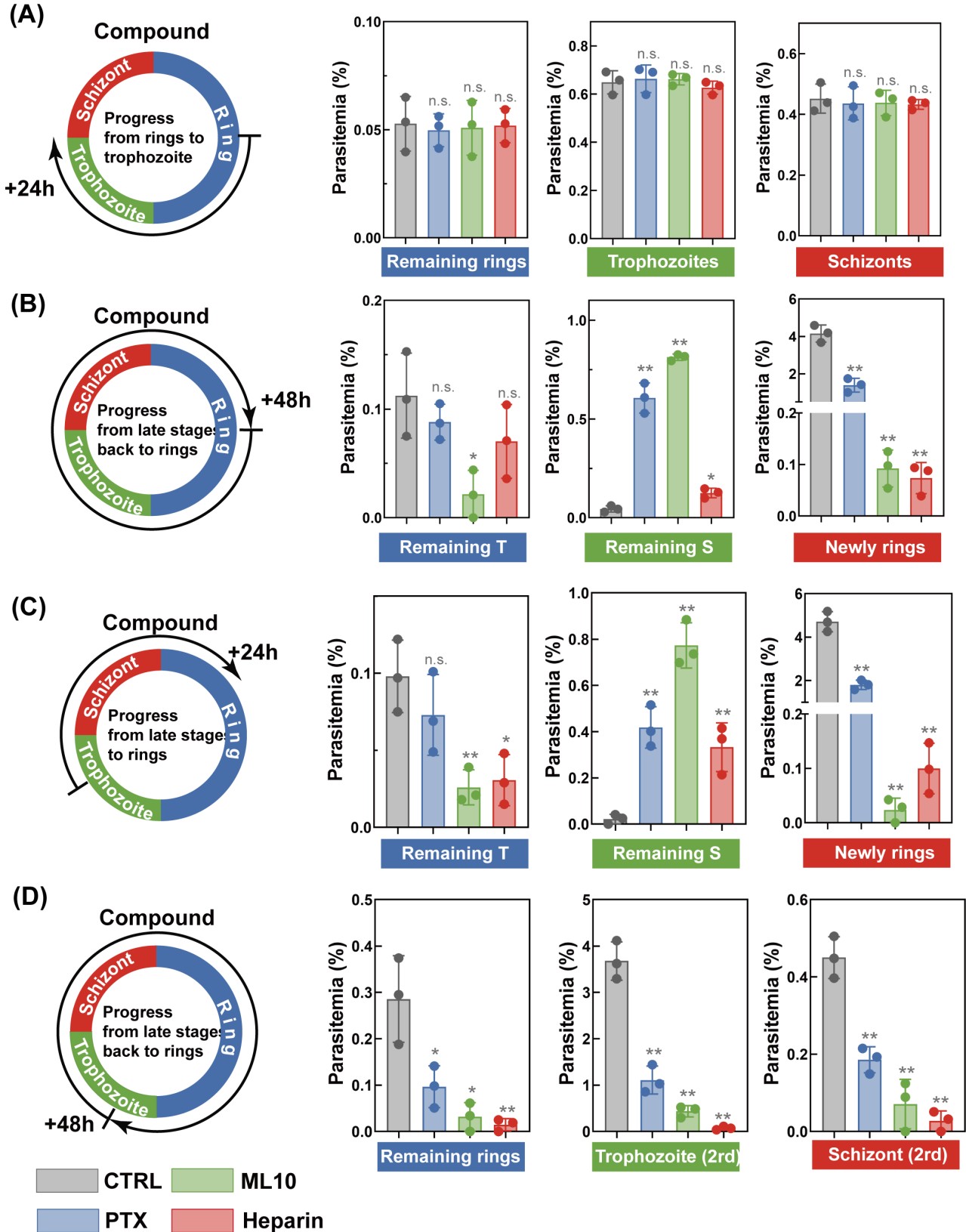

**FIG 4** Long-term exposure to PTX impairs late-stage parasite development through IDC. To assess PTX's impact on stage-specific developmental progression of *P. falciparum*, (A and B) synchronized ring-stage parasites and (C and D) synchronized trophozoite-stage parasites were treated with PTX at a concentration of $1\times IC_{50}$ for the indicated time periods. Following drug exposure, parasite developmental progression was monitored via microscopic analysis of Giemsa-stained

Fig 4 (Continued)

smears. Parasite progression after exposure was microscopically monitored by Giemsa-stained smears with at least 5,000 erythrocytes counted per sample to quantify ring, trophozoite, and schizont stages. ML10 (200 nM), a validated egress inhibitor, served as a control to confirm PTX's effect on parasite egress from erythrocytes. Heparin (100 µg/mL), a validated invasion inhibitor blocking erythrocyte receptor engagement, was used as a control to validate inhibition of merozoite invasion. Data were presented as mean ± SD from independent experiments with technical triplicates. Statistical significance was analyzed via one-way ANOVA followed by Bonferroni post hoc correction *$P < 0.05$, **$P < 0.01$, and n.s. indicating no significant difference.

simulations identified three key structural perturbations triggered by PTX binding to the β-subunit of PfTubulin. First, the RMSD of the PTX-PfTubulin complex fluctuated within the range of 2.2–6.4 Å during the initial 200 ns. This phase reflects the exploration of the binding pocket by the ligand. The RMSD then stabilized at approximately 5 Å for the subsequent 800 ns (Fig. 6B). This profile indicates a thermodynamically favorable interaction consistent with PTX's function in locking tubulin into polymerized states. This RMSD pattern is consistent with findings from MD studies of PTX-human tubulin complexes, where an RMSD of approximately 4–6 Å was linked to microtubule stabilization (71).

Additionally, RMSF analysis suggested a marked increase in flexibility with RMSF values ranging from approximately 1.8–2.5 Å across PfTubulin residues 430–444 (Fig. 6C). This region corresponds to the MAP-interacting M-loop, a domain critical for longitudinal tubulin heterodimer polymerization and microtubule lattice integrity (72). Enhanced flexibility in this region may disrupt adjacent subunit interactions during schizont microtubule assembly, a biological process essential for nuclear segregation and merozoite biogenesis (73). This observation aligns with cryo-EM findings reporting irregular polymerization in PTX-bound microtubules (74). Notably, the Rg of PfTubulin decreased from approximately 2.2 nm in the apo state to approximately 2.1 nm following PTX binding (Fig. 6D). This change is indicative of global structural compaction that may alter the tubulin binding interface and impair lateral protofilament interactions, which is a key step in microtubule assembly (23). Such structural compaction may further undermine the formation of mitotic spindles and subpellicular microtubules during schizogony (28).

These PTX-induced structural perturbations may contribute to the observed phenotypic defects of *P. falciparum*. Schizogony arrest depends on coordinated microtubule dynamics, including spindle formation for nuclear division and subpellicular microtubules for cell shape remodeling. PTX-stabilized tubulin with low RMSD values and enhanced M-loop flexibility may disrupt spindle pole organization and lead to failed nuclear segregation. This molecular mechanism is consistent with experimental observations of PTX-treated schizonts showing abnormal nuclear lobing and delayed merozoite release (Fig. 2B through E) and aligns with phenotypic changes induced by other microtubule stabilizers. For reduced merozoite formation, a process that requires microtubule-mediated trafficking of apical complex components such as rhoptries and micronemes to the cell periphery, compacted PfTubulin with a lower Rg value may impair the binding of kinesin motors, including PfKinesin-8, thereby blocking organelle transport (75). Experimentally, PTX treatment reduced merozoite yield by approximately 32% from 19 to 13 per schizont and increased the proportion of unruptured schizonts by fourfold (Fig. 2C and F). This observation is in line with findings in *Toxoplasma gondii*, where PTX-induced microtubule compaction diminishes microneme secretion and host cell invasion (22).

In conclusion, MD simulations support a plausible mechanistic link between PTX's molecular interaction with PfTubulin and its resultant phenotypic effects on *P. falciparum*. Specifically, stabilized PTX-PfTubulin binding enhanced M-loop flexibility and global structural compaction may act in concert to disrupt the coordinated microtubule dynamics essential for parasite development. These perturbations are proposed to drive schizogony arrest and reduced merozoite formation, thereby providing support for the core antimalarial mechanism of PTX. The data further identify the PfTubulin M-loop as a

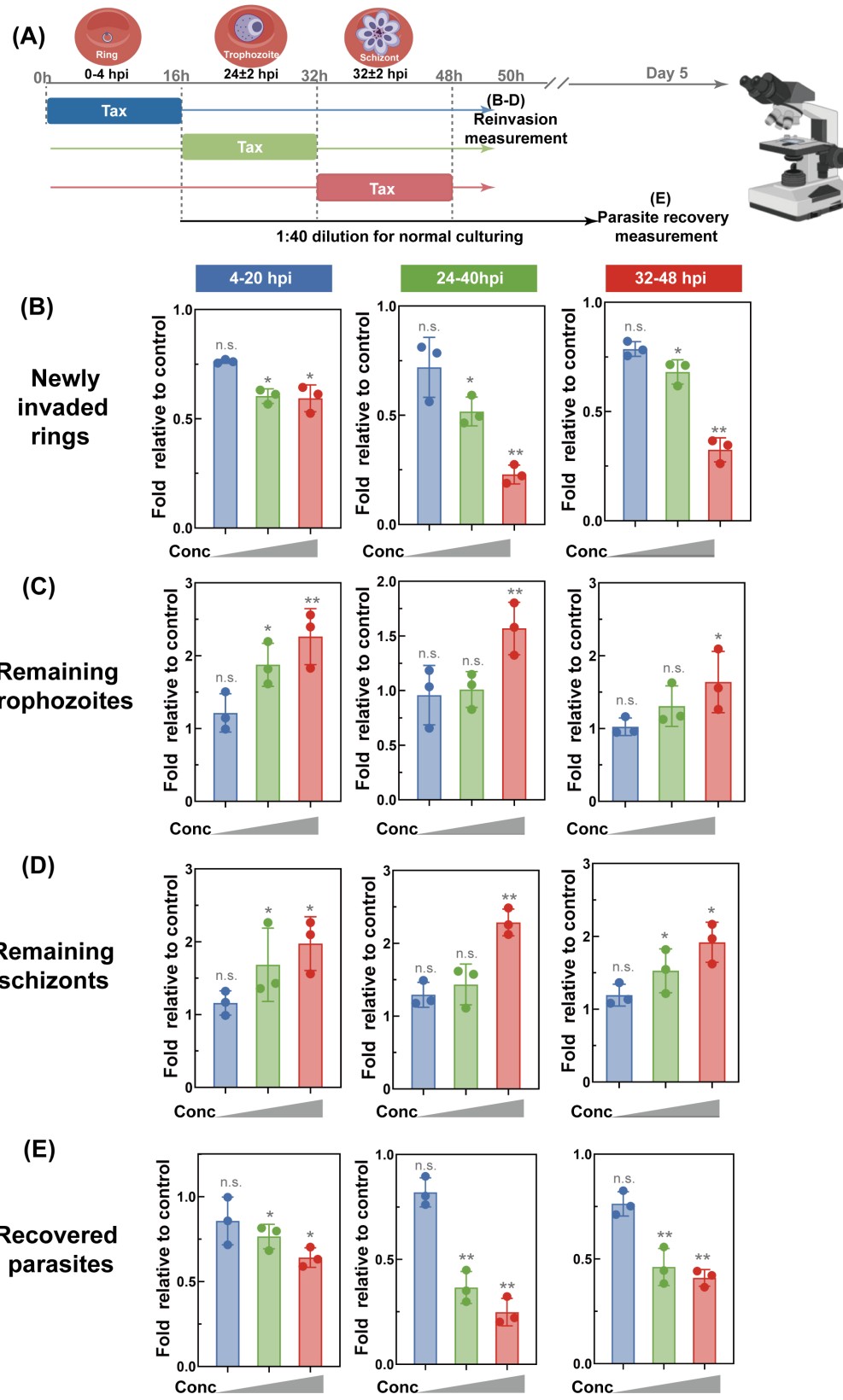

FIG 5   Transient exposure to PTX disrupts development at all stages throughout the IDC with long-term consequences.
(A) Schematic of the experimental design in which highly synchronized parasites were exposed to the vehicle or 100, 500, or
1,000 nM PTX (corresponding to 1×, 5×, or 10× IC$_{50}$) during one of three 16-h periods: ring (0–4 hpi), trophozoite (22–26 hpi),

**Fig 5 (Continued)**

or schizont (30–34 hpi). Following treatment, parasite developmental progression through the IDC and subsequent reinvasion efficiency were microscopically evaluated at 50 hpi. (B–D) Quantification of parasite stage distribution post-PTX exposure. (B) Fold change in newly invaded ring-stage parasites, (C) fold change in remaining trophozoite-stage parasites, and (D) fold change in schizont-stage parasites that failed to complete the IDC, all normalized to vehicle-treated control parasites. These metrics reflect the stage-specific inhibitory effects of PTX on parasite development and egress from infected erythrocytes. (E) Long-term parasite survival assay following transient PTX exposure. At the conclusion of each 16-h treatment period, parasites were pelleted by centrifugation, washed with fresh culture medium to remove residual drug, and diluted 1:40 to initiate recovery culture. Parasite cultures were maintained for two additional complete IDC cycles, and surviving parasites were microscopically quantified on day 5 post-treatment. Data were presented as mean ± SD from independent experiments with technical triplicates. Statistical significance was analyzed via one-way ANOVA followed by Bonferroni post hoc correction *P < 0.05, **P < 0.01, and n.s. indicating no significant difference.

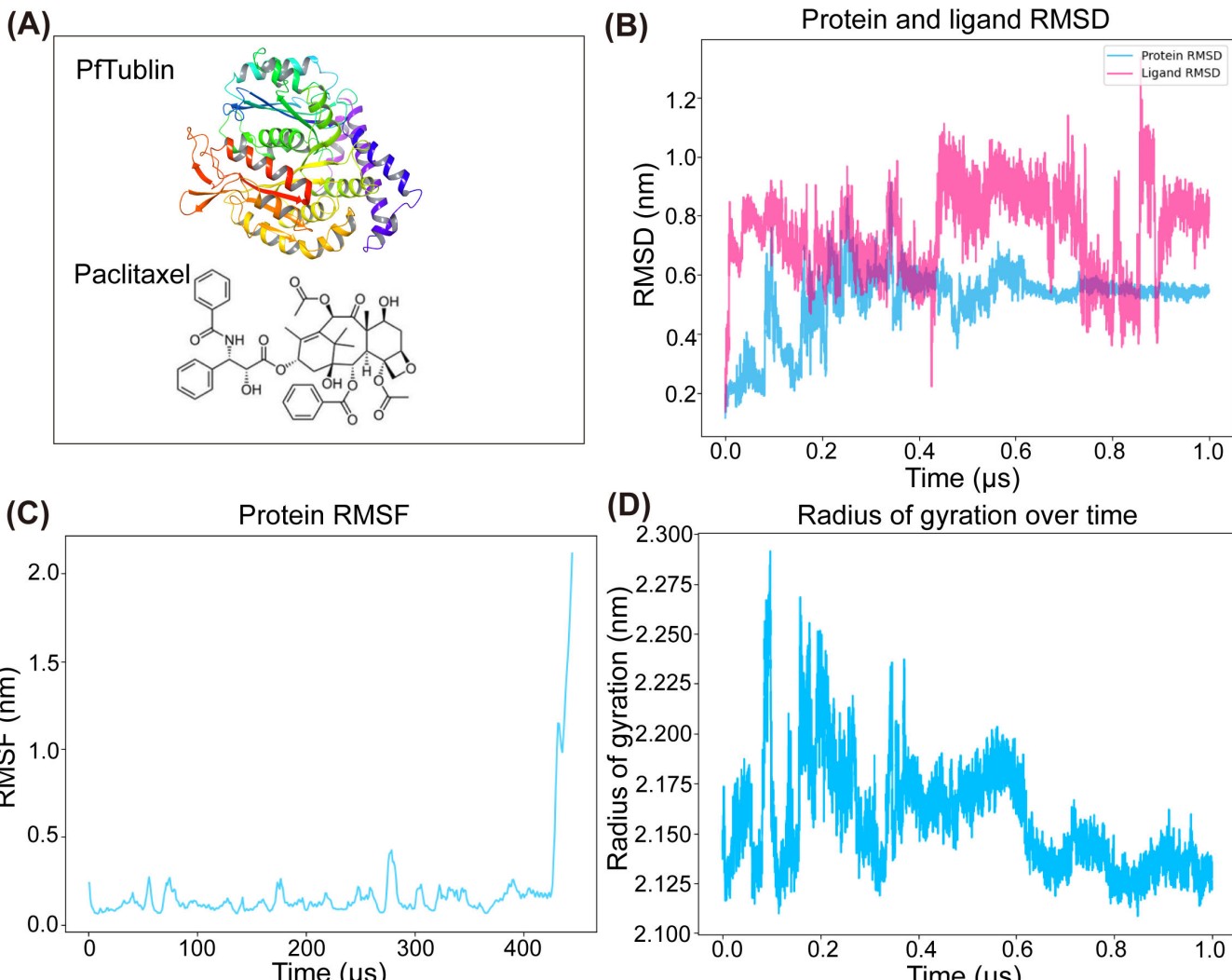

**FIG 6** Molecular dynamics simulation of the interaction between PTX and PfTublin. (A) 3D structural model of PfTublin and the molecular structure of PTX. (B) RMSD analysis from 1,000 ns MD simulations of the PfTubulin-PTX complex. For the simulation system, a grid box was centered at coordinates (−15.05, −12.3, and 2.44) Å around the predicted PTX-binding pocket of PfTubulin, with dimensions set to 20 × 20 × 20 Å to ensure comprehensive sampling of the binding interface and surrounding structural regions. The RMSD plot reflects the structural stability of the PfTubulin-PTX complex over the course of the simulation. (C) Root-mean-square fluctuation (RMSF) analysis of PfTubulin residues during the 1,000 ns MD simulation. This plot quantifies the flexibility of individual amino acid residues in PfTubulin following PTX binding, highlighting regions of the protein that undergo significant conformational changes. (D) Radius of gyration (Rg) analysis to characterize global structural changes of PfTubulin. The Rg plot visualizes variations in the compactness of the PfTubulin structure throughout the physiological MD simulation, providing insights into PTX-induced changes in the protein's overall conformation.

promising molecular target for the rational design and optimization of next-generation antimalarial PTX derivatives.

## Synergistic antimalarial activity of PTX in combination with conventional antimalarials

To determine whether the co-application of PTX and traditional antimalarials could enhance therapeutic efficacy against drug-resistant parasite strains, a combination assay was performed where parasite strains were exposed to individual drugs or predefined fixed-ratio combinations. Specifically, parasite viability was assessed using a 72-h SYBR-Green I assay following treatment with DHA + PTX, CQ + PTX, or PYR + PTX combinations at concentrations of $0.25\times$ $IC_{50}$, $1\times$ $IC_{50}$, $2\times$ $IC_{50}$, and $4\times$ $IC_{50}$ (Fig. 7A). By analyzing parasite viability and calculating the CI using the Chou-Talalay method, the results revealed that most DHA- or CQ-PTX combinations exhibited enhanced antimalarial activity. The DHA-PTX combination exhibited synergistic activity at sub-$IC_{50}$ concentrations, with CI values ranging from 0.802 to 0.954, which was indicative of mild to moderate synergism against the parasite strains (46). Additionally, the CQ + PTX combination exhibited synergistic activity at $0.5\times$ and $1\times$ $IC_{50}$ concentrations (Fig. 7B). Collectively, these findings highlight a synergistic antimalarial activity in combinatorial therapeutic efficacy, positioning PTX as a promising partner drug for overcoming emerging drug resistance in malaria parasites.

Based on the previously validated *in vitro* synergistic antimalarial activity, our subsequent objective was to investigate whether the co-administration of PTX with DHA could effectively contribute to partially overcoming artemisinin resistance. To this end, both standard and slightly modified $RSA_{0-3\,h}$ assays were performed to quantify survival rates of resistant parasite strains; the experimental scheme is depicted in Fig. 7C. This assay was originally developed to assess delayed parasite clearance in artemisinin-resistant isolates by measuring parasite survival after a 6-h drug exposure, which mirrors the clinical hallmark of artemisinin resistance (49). As shown in Fig. 7D, the single use of the compounds did not significantly affect the survival rate, potentially due to the limited exposure period, which was insufficient for PTX to exert its full effects. This finding aligns with prior studies showing that mono-therapies with slow-acting antimalarials often fail to suppress resistant parasites within short exposure windows (13, 76). To further elucidate the post-incubation effect and stage-specific antimalarial activity, we conducted a time and stage-dependent assay. Parasites were exposed to the PTX at a dosage of $1\times$ $IC_{50}$ for 16 h during distinct developmental stages, ring, trophozoite, and schizont, followed by the standard $RSA_{0-3\,h}$ protocol. In accordance with the stage-specific assay results, co-treatment with the PTX significantly enhanced the clearance of DHA-surviving parasites. These findings corroborate prior research showing that PTX can disrupt the transition between parasite life-cycle stages (77), thereby increasing parasite vulnerability to artemisinin. The observed enhancement in DHA efficacy, especially against late-stage parasites, has significant implications for malaria treatment.

Given the preliminary findings that combination treatment effectively suppresses the survival of artemisinin-resistant *Plasmodium falciparum* strains (6, 78), we further investigated the concentration-dependent efficacy of the PTX-DHA combination. As illustrated in Fig. 7F and G, parasites exposed to the PTX-DHA combination exhibited a significantly lower survival rate compared to those treated with either compound alone. After a 6-h exposure to DHA (fixed at 700 nM) and PTX (at varying concentrations, 25–200 nM), followed by a 66-h recovery period, ring-stage survival in the combination group dropped below 5%, and a dose-dependent decrease in survival was observed as PTX concentration increased. In contrast, DHA monotherapy resulted in survival rates of 10% and 7% for the $K13^{C580Y}$ (Fig. 7F) and $K13^{G538V}$ (Fig. 7G) mutant strains, respectively. These survival rates in DHA monotherapy were markedly higher than those observed in the PTX-DHA combination treatment group, and this elevated parasite survival indicated reduced antiplasmodial efficacy against these artemisinin-resistant strains. This notable difference further underscored the enhanced antiplasmodial activity afforded by the

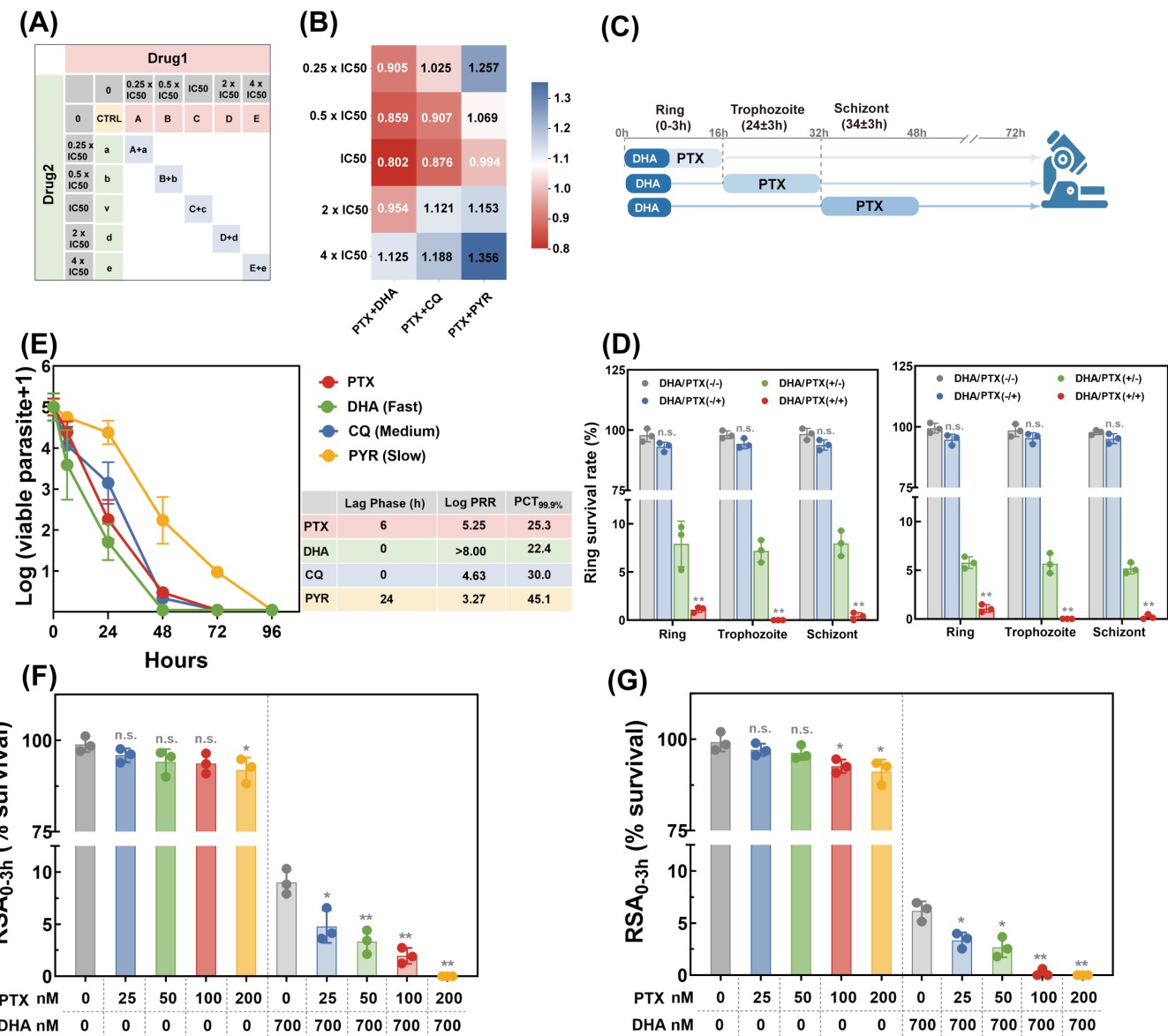

**FIG 7** Synergy effect and mode of action of PTX against *P. falciparum*. (A) Schematic representation of the design of the drug combination experiment. Drug 1 corresponds to PTX, and drug 2 corresponds to typical antimalarials, including DHA, CQ, and PYR. Parasites were treated with either drug alone or in combination with a fixed ratio for 72 h, and the parasite viability was measured by SYBR-Green I assay. The combination index was calculated using the Chou-Talalay method, in which CI < 1 corresponds to a synergy effect, while CI > 1 corresponds to an antagonistic effect. (B) Synergy heat map depicting CI values between PTX and DHA/CQ/PYR against 3D7 WT strain at the previously mentioned combinations. (C) Schematic overview of experimental design for co-incubated $RSA_{0-3}$ h. Highly synchronized parasites with ~3 h window were subjected to co-treatment between 700 nM DHA and PTX at ring/trophozoite/schizont stage for 6 h and 12 h, respectively, as shown. (D) Survival rate for K13$^{C580Y}$ (left panel) and K13$^{G538V}$ (right panel) parasite strains at 72 h was microscopically assessed based on the parasitemia examined from vehicle control and drug-treated samples. Parasitemia was calculated from at least 10,000 RBCs by microscopy using Giemsa-stained smears. (E) Parasite viability time course in response to traditional antimalarials and PTX in which correlates with their mode of action. Viability time-course profiles of DHA, CQ, PYR, and PTX illustrating the wide range of speed-of-action measured. Key parameters obtained from the parasite viability measurement, including drug lag phase, parasite reduction ratio (PRR), and 99% parasite clearance time (PCT) were listed in the inset table. In detail, the lag phase corresponded to the time required for reaching the maximal rate of killing, PRR corresponded to the ratio between parasitemia at the onset of drug treatment and after one life cycle, 99.9% PCT corresponded to the time required to decrease the number of viable parasites by 3-log units. Survival rate of resistant parasite lines K13$^{C580Y}$ (F) and K13$^{G538V}$ (G) subjected to either single or co-exposure of DHA and PTX at 1/4×, 1/2×, 1×, and 2× IC$_{50}$ followed by standard $RSA_{0-3h}$. Parasite survival rate was calculated as described. Data were presented as mean ± SD from independent experiments with technical triplicates. Statistical significance was analyzed using two-tailed student's t-test *P < 0.05, **P < 0.01, and n.s. indicating no significant difference.

combinatorial regimen against such resistant parasites. Notably, PTX monotherapy showed negligible antimalarial activity, with parasite survival exceeding 95% under identical conditions. These results underscore the critical role of concentration ratios in combination therapies for overcoming artemisinin resistance. PTX-DHA synergy likely stems from complementary targeting of distinct metabolic pathways: DHA's rapid ROS-mediated damage to early rings synergizes with PTX's interference with tubulin. It has been reported that reactive oxygen species (ROS) impair cytoskeletal integrity by directly damaging microtubule structure and disrupting its dynamic balance (79–81). Multiple studies confirm that ROS induce the oxidation or nitration of key amino acid residues in tubulin, resulting in microtubule lattice loosening, fragmentation, or the formation of abnormal polymers (82–85). Meanwhile, ROS inhibit the function of end-binding protein 1 (EB1), a critical microtubule plus-end binding protein, and disrupt the dynamic stability of microtubule polymerization-depolymerization cycles (86). This cascade of events ultimately leads to a disorganized, functionally compromised microtubule network.

As a classic tubulin stabilizer, the core of PTX's synergistic effect with ROS is proposed to not involve random stabilization but rather to specifically lock these ROS-induced abnormal microtubule structures. Research indicates that PTX binds specifically to a conserved site on the β-subunit of tubulin. This interaction is thought to drive the polymerization of tubulin heterodimers while potently suppressing microtubule depolymerization. Notably, this stabilizing activity appears to be non-discriminatory. Even when ROS have already caused structural damage to microtubules such as lattice defects, amino acid modifications, or partial fragmentation, PTX may still trap these defective microtubules in a rigid, stable state. This potential trapping effect could be biologically significant because it might prevent parasites from activating their intrinsic adaptive repair mechanisms. Normal cells typically clear impaired microtubule segments through depolymerization. They then reorganize functional tubulin subunits to restore cytoskeletal order. PTX-mediated inhibition of microtubule depolymerization might affect this repair process. The locked abnormal microtubules may not only persist as structural anomalies but also fail to participate in essential cellular processes, and this may potentially further amplify the functional impairment of the cytoskeleton initiated by ROS. Collectively, our data emphasize the therapeutic potential of PTX as a partner drug to DHA, particularly in combating artemisinin resistance in early-stage parasites.

## *In vitro* evaluation of killing kinetics for PTX

To comprehensively assess the net impact of PTX on parasite viability, an *in vitro* PRR assay was meticulously conducted. This assay is a well-established method for quantifying the number of parasites that retain the ability to recrudesce following drug removal, offering critical insights into drug efficacy. By implementing iterative limited serial dilution procedures, the time-dependent changes in parasite viability in response to PTX exposure were systematically monitored. This approach allowed for the precise determination of key *in vitro* parasite killing rates, which were calculated based on PCT, PRR values, and lag phases, key parameters of paramount importance in antimalarial drug evaluation. The PRR and PCT are directly correlated with the *in vivo* parasite elimination kinetics, playing a pivotal role in determining patient symptom resolution and treatment cure rates. A short PCT is a fundamental objective in antimalarial drug design, as rapidly acting drugs significantly reduce the exposure time of parasites to suboptimal drug concentrations (47). This reduction minimizes the selective pressure for resistant mutant emergence, a critical factor in maintaining the efficacy of antimalarial therapies. Additionally, the lag phase provides valuable information regarding the time required for a drug to initiate its mechanism of action. A prolonged lag phase can lead to delayed parasite death, thereby increasing the risk of clinical treatment failure. The viability-time curve generated from the study demonstrated that the number of viable parasites decreased to less than 0.01% of the initial count (Fig. 7E). This result strongly indicated that the chosen incubation period was sufficient to fully capture the activity of the tested compounds. Among the drugs evaluated, the slow-acting pyrimethamine

(PYR) exhibited a pronounced 24-h lag phase, with a log (PRR) value of 3.27 and a 99.9% PCT of 45.1 h, consistent with previous findings (87). Dihydroartemisinin (DHA), chloroquine (CQ), and PTX followed in descending order of action speed, with log PRR values of >8.00, 5.25, and 4.63, respectively. The PRR assay results after a 28-day recovery period revealed that PTX exerted its antimalarial effect rapidly, with a performance comparable to that of CQ, completely eradicating all parasites within 24 h of exposure. The killing rate profiles further illustrated the significant difference in efficacy between fast-acting and slow-acting antimalarials. Fast-acting drugs such as DHA, CQ, and PTX were able to clear viable parasites within 72 h, whereas slow-acting agents, including PYR, failed to achieve complete clearance within the same time frame. The escalating problem of parasite resistance to existing antimalarials underscores the urgent necessity of discovering and developing novel fast-acting drugs (88). This is crucial for preventing potential failures of artemisinin-based combination therapies (ACTs), the current gold standard for malaria treatment (6). *In vitro* evaluation of killing rate parameters, as demonstrated in this study, provides essential preclinical data for the screening and development of new antimalarial candidates. These parameters not only serve as predictors of *in vivo* efficacy but are also closely associated with the drugs' MOA. For instance, drugs with different MOAs may exhibit distinct patterns of PRR, PCT, and lag phase, which can guide the rational design of new antimalarials.

### *In vivo* combinational antimalarial activity of PTX-DHA in rodent malarial model

To comprehensively assess the *in vivo* antimalarial efficacy of PTX, parasite clearance profiles were systematically evaluated in rodent malaria model infected with either drug-sensitive or -resistant *P. berghei* ANKA strain. PTX was administered to the infected mice at low (10 mg/kg), medium (20 mg/kg), and high (30 mg/kg) concentrations, along with vehicle controls, and the parasitemia levels were continuously monitored over time. In the untreated control group, parasitemia exhibited a consistent upward trend, reaching 21.6% ± 5.6% by day 15 post-infection, which is indicative of severe malaria progression (Fig. 8A). A clear concentration-dependent effect of PTX was observed, in which a pronounced antimalarial effect was achieved with increased PTX concentration. The parasitemia levels decreased by 81.5% ± 11.5%, 88.5% ± 4.0%, and 96.1% ± 1.2% for each group compared to the control group at day 5 (Fig. 8E). Although this reduction was not sufficient to completely clear the parasite, survival analysis revealed a notable improvement in the survival rate. Kaplan–Meier survival curves demonstrated that the median survival time was extended compared to untreated counterparts (Fig. 8I). Subsequently, the *in vivo* synergistic effect of the antimalarial compounds was evaluated. In the group treated with DHA monotherapy at a dose of 5 mg/kg, parasitemia growth was notably retarded, reaching 2.0% ± 0.4% by day 10, and the inhibition rate was 84.8% ± 4.7% at day 10 (Fig. 8B and F). Meanwhile, the combination-treated group exhibited a significantly lower parasitemia level, with values bottoming to 0.09% ± 0.05% by day 10 with nearly completed parasite clearance at the inhibition rate of 97.7% ± 0.6% (Fig. 8B and F). Remarkably, all mice in the combination group survived throughout the assay, while the survival rates for the DHA-treated group only reached 60% (Fig. 8J).

The therapeutic potential of PTX in combination with DHA was further evaluated in an artemisinin-resistant *P. berghei*-infected rodent model. Similar to the above-mentioned findings against a drug-sensitive strain, PTX at 10/20/30 mg/kg concentrations demonstrated a concentration-dependent antimalarial effect. By day 5 post-infection, parasitemia levels in the PTX-treated groups decreased by 85.3% ± 2.4%, 93.3% ± 2.7%, and 96.8% ± 1.5% compared to the vehicle-control group. Despite the inability to achieve complete parasite clearance, survival analysis using Kaplan–Meier curves revealed a substantial improvement in the median survival period of treated mice. The median survival time extended from 20% (control) to 60% in the PTX-treated groups (Fig. 8C, G, and K). In combination studies, DHA monotherapy at a dose of 5 mg/kg also exhibited antimalarial activity, reducing parasitemia to 0.4% ± 0.1% by day 10,

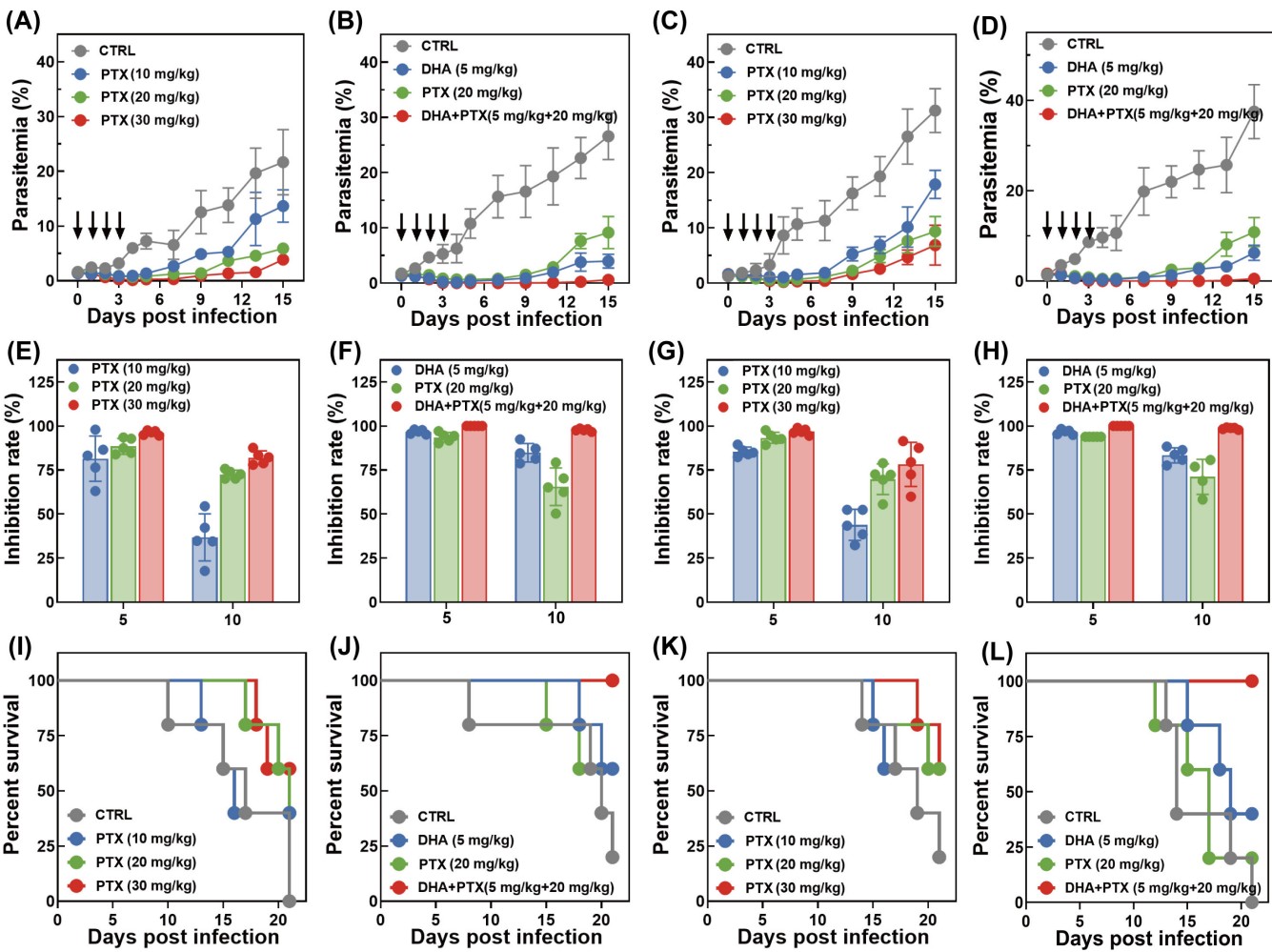

**FIG 8** *In vivo* antimalarial activity evaluation of PTX in rodent model against both drug-sensitive and resistant *P. berghei* strains. *In vivo* activity against the blood stages of mice infected with *P. berghei* drug-sensitive strain by single-drug (A) or combination treatment (B). *In vivo* activity against the blood stages of mice infected with *P. berghei* artemisinin-resistant strain by single-drug (C) or combination treatment (D). Drug treatment was intraperitoneally administered for four consecutive days (indicated by the arrow) after infection according to the standard Peters' inhibition assay protocol. (E and F) Inhibition rate for mice post artemisinin-sensitive *P. berghei* strain infection upon administration with single (E) or combination treatment (F) at day 5 and day 10. Inhibition rate for mice post-artemisinin-resistant *P. berghei* strain infection upon administration with single (G) or combination treatment (H) at day 5 and day 10. (I–L) Kaplan–Meier survival curve showing the survival of *P. berghei*–infected mice from the experiment described in the panel for a period of 21 days. Five mice were used in each group, and parasitemia was recorded daily and calculated from at least 5,000 RBCs by microscopy.

corresponding to an inhibition rate of 96.6% ± 1.2% with 40% survival during the observation period. In contrast, the PTX-DHA combination achieved remarkable results. By day 5 post-infection, the mean parasitemia dropped significantly to 0%, demonstrating a rapid and potent antimalarial effect consistent with the complementary action of PTX and DHA (Fig. 8D, H, and L). Identical trends were observed in drug-resistant strains, reaffirming the broad-spectrum efficacy of the PTX-DHA combination against both sensitive and resistant parasites.

## Therapeutic potential of PTX as a partner drug by inhibiting the plasmodium growth and recrudescence

The *in vitro* recrudescence assay for antimalarial drugs serves as a pivotal tool in malaria research, enabling the simulation of the *in vivo* infection and recrudescence processes under controlled laboratory conditions. In this study, parasites were exposed to DHA alone, PTX alone, or the drug combinations, followed by multiple drug administrations

over a 28-day observation period. Parasitemia was microscopically quantified at regular intervals to monitor recrudescence, mirroring the clinical situation of post-treatment relapse. The results revealed that the group treated with DHA alone exhibited a rapid resurgence of parasitemia after an initial decline, regardless of the treatment duration or the resistance level of parasite strains (Fig. 9A through C). This finding aligns with previous reports highlighting the short-lived effect of DHA against persistent parasites, leading to early recrudescence (51, 89). In K13$^{C580Y}$ parasite lines, parasitemia reached 2.5% ± 0.4%, 3.4% ± 0.4%, and 2.6% ± 0.3% following 6-h, 9-h, and 12-h treatments with DHA, respectively (Fig. 9A through C). Similarly, in K13$^{G538V}$ parasite lines, parasitemia values were 3.7% ± 0.5%, 1.4% ± 0.2%, and 1.7% ± 0.3% after 6-h, 9-h, and 12-h DHA treatments. Notably, the combination of PTX and DHA significantly suppressed recrudescence, with no recrudescence observed in parasites subjected to 12-h exposure throughout the entire observation period (Fig. 9C). The complementary action of PTX, which targets tubulin, and DHA, which induces rapid oxidative stress, likely disrupts multiple survival mechanisms in the parasite, preventing its resurgence. The combination of PTX and DHA shows great promise in preventing malaria recrudescence, highlighting the potential of this regimen in improving clinical outcomes for malaria patients, especially those infected with artemisinin-resistant strains.

## Synergistic antimalarial efficacy of PTX-DHA in humanized mice model against *P. falciparum*

While rodent malarias, such as *P. berghei* and *P. yoelii*, have long served as a cornerstone for initial antimalarial compound screening due to their genetic tractability and rapid lifecycle, their limited translational predictability necessitates advanced models. As highlighted, only 12% of compounds demonstrating efficacy in rodent models progress to successful human trials, underscoring the critical need for more physiologically relevant systems (90). To bridge this translational gap, a humanized mouse model engrafted with human erythrocytes was employed to evaluate the *in vivo* activity of PTX against *P. falciparum*. NCG-X mice were intravenously engrafted with $2 \times 10^8$ human erythrocytes, following a protocol adapted from previous studies. Daily flow cytometry analysis of CD235a-stained peripheral blood samples was performed to monitor the engraftment kinetics. Once the human erythrocyte engraftment stabilized at approximately 40% (Fig. 9D through F), the mice were inoculated with $1 \times 10^7$ parasitized human erythrocytes (3D7-sensitive strain or artemisinin-resistant strain). During the 4-day treatment course, the PTX-DHA combination therapy effectively suppressed parasite growth, maintaining parasitemia below 0.1% at day 10 post-infection, superior to DHA monotherapy (Fig. 9G through I). Notably, the DHA-PTX combination achieved a 98.2%, 96.5%, and 96.9% reduction in parasitemia and induced a substantial delay in parasite recrudescence across both sensitive and resistant strains. As *P. falciparum* selectively targets human erythrocytes, and traditional rodent models often fall short in mimicking this host–parasite interaction accurately due to species-specific differences, this humanized model, in which reconstituted with human-derived erythrocytes, provides a more physiologically relevant model. These models support the complete erythrocytic cycle of *P. falciparum*, enabling researchers to study parasite invasion, growth, and replication within human-like cellular environments. In drug-resistant strain studies, these models have shown consistent responses to combination therapies like PTX-DHA, mirroring human-relevant resistance mechanisms.

## Conclusion

To investigate the antimalarial mechanism of PTX and its potential as a combinatorial partner for artemisinin, this study employed an integrated suite of pharmacological and biochemical approaches. In this study, we have demonstrated that PTX disrupts *Plasmodium* intraerythrocytic development through microtubule stabilization, a mechanism validated by its inhibition of tubulin depolymerization. Our *in vitro* and *in vivo* data show that the resultant disruption of microtubule dynamics induced by PTX

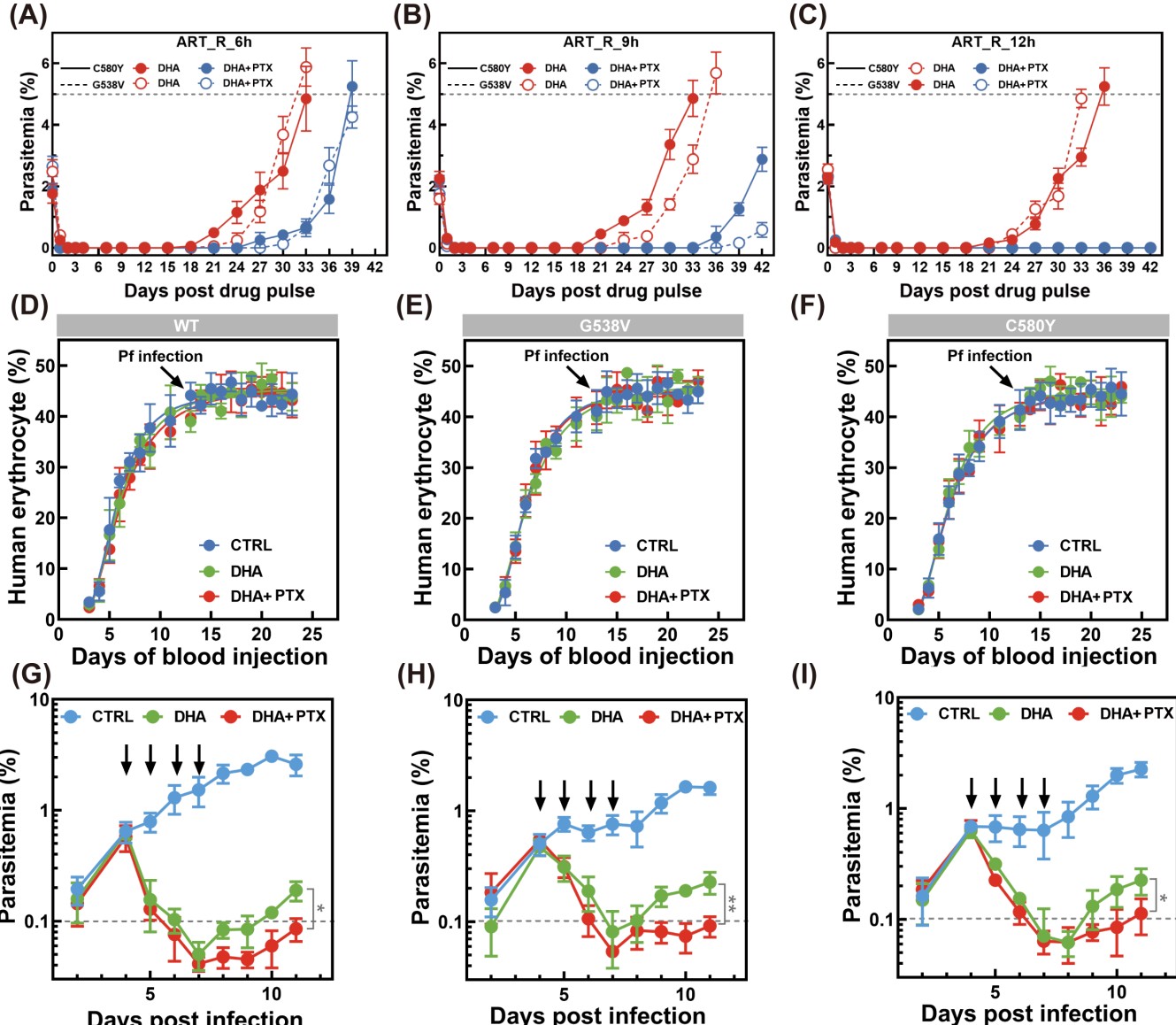

**FIG 9** *In vitro* and *in vivo* recrudescence evaluation of PTX against both drug-sensitive and resistant *P. falciparum* strains. (A–C) Recrudescence assay of artemisinin (ART)-resistant *P. falciparum* strains following repeated drug exposure. Ring-stage parasites were subjected to five sequential drug pulses (each lasting 6, 9, or 12 h) with either single-drug or combination treatment. Recrudescence curves were generated based on parasitemia levels quantified from Giemsa-stained blood smears, reflecting the ability of resistant parasites to recover and regrow after repeated drug pressure. (D–F) Assessment of the engraftment efficiency of the humanized model by measuring the percentage of human erythrocytes in peripheral blood. Mice with stable peripheral hRBC levels (~40%) were infected with different *P. falciparum* strains at the time point illustrated by the arrow. (G–I) *In vivo* antimalarial activity kinetics profiles in humanized mice. Peripheral blood parasitemia levels were monitored daily to evaluate the efficacy of treatments against three *P. falciparum* strains: wild-type (WT, G), K13_G538V mutant (H), and K13_C580Y mutant (I) strains. Three mice were used in each group, and parasitemia was recorded daily and calculated from at least 10,000 RBCs by microscopy. Statistical significance was analyzed via one-way ANOVA followed by Bonferroni post hoc correction *$P < 0.05$, **$P < 0.01$.

exposure is lethal to the parasite, leading to arrested growth, impaired mitosis, and reduced viability. Although PTX exhibited promising antimalarial potential in our experimental findings, it should also be worth noting that its inherent cytotoxicity should never be ignored. A validated strategy to mitigate such cytotoxicity is the encapsulation of PTX into biocompatible nanocarriers (e.g., albumin nanoparticles), which can effectively improve targeted delivery and reduce off-target toxic effects on normal somatic cells. In this study, we primarily aimed to explore PTX's potential as a targeted antimalarial agent; meanwhile, in consideration of its unavoidable cytotoxicity,

we further sought to verify whether it could act as a synergistic partner drug in combination with ART. Therefore, we placed greater emphasis on the co-administration of PTX and DHA in both *in vitro* and *in vivo* assays and systematically analyzed their combined antimalarial efficacy and safety characteristics. Novel combinatorial potential with DHA was validated in both parasite reduction ratio assays and erythrocyte-humanized mice models, and PTX-DHA combinations exert multi-targeted effects on the parasite. This combinatorial approach has enhanced treatment efficacy, as evidenced by significantly reduced parasitemia and delayed recrudescence. In addition, structural modification of PTX has also been carried out, and we have successfully synthesized several PTX derivatives based on its core molecular structure. Further in-depth efforts will be made to comprehensively evaluate the antimalarial activity, cytotoxicity, and structure–activity relationship of these derivatives, so as to develop safer and more effective antimalarial candidates derived from PTX. In summary, PTX's ability to inhibit *Plasmodium* growth during the intraerythrocytic asexual stage via microtubule-mediated mechanisms establishes it as a promising candidate for novel malaria therapies. Beyond its therapeutic potential, this discovery sheds light on parasite cytoskeletal biology and provides critical insights for future antimalarial drug development, including the design of novel combination regimens and the exploration of alternative targets.

## ACKNOWLEDGMENTS

This work was supported by the National Natural Science Foundation of China (82320108014 and 82102424) and sponsored by the TaihuLight Science and Technology Research Project.

## AUTHOR AFFILIATIONS

[1]Center for Global Health, School of Public Health, Nanjing Medical University, Nanjing, China

[2]National Health Commission Key Laboratory of Parasitic Disease Control and Prevention, Jiangsu Provincial Key Laboratory on Parasite and Vector Control Technology, Jiangsu Institute of Parasitic Diseases, Wuxi, China

[3]State Key Laboratory of Biomacromolecules, Institute of Biophysics, Chinese Academy of Sciences, Beijing, China

## AUTHOR ORCIDs

Yongxin Tang  http://orcid.org/0009-0003-2252-2539
Tao Jiang  http://orcid.org/0000-0003-1427-4793
Jun Cao  http://orcid.org/0000-0001-5298-9234
Xinyu Yu  http://orcid.org/0000-0002-1339-3558

## FUNDING

| Funder | Grant(s) | Author(s) |
| --- | --- | --- |
| National Natural Science Foundation of China | 82102424 | Xinyu Yu |
| National Natural Science Foundation of China | 82320108014 | Jun Cao |

## AUTHOR CONTRIBUTIONS

Yongxin Tang, Investigation, Methodology, Software, Validation, Writing – original draft | Xinyu Zhang, Investigation, Methodology, Visualization | Xiaohui He, Formal analysis, Methodology, Validation | Tao Jiang, Conceptualization, Data curation, Writing – original draft | Jun Cao, Conceptualization, Formal analysis, Funding acquisition, Validation, Writing – review and editing | Xinyu Yu, Conceptualization, Data curation, Formal analysis, Funding acquisition, Writing – review and editing

## ETHICS APPROVAL

This study was carried out in accordance with the institutional guidelines for the care and use of animals at Jiangsu Institute of Parasitic Diseases (JIPD). The experimental design was reviewed and approved by the Research Ethics Committee of JIPD (JIPD-2023-003, JIPD-2023-014).

## ADDITIONAL FILES

The following material is available online.

### Supplemental Material

**Supplemental material (Spectrum01957-25-s0001.docx).** Fig. S1; Table S1.

### Open Peer Review

**PEER REVIEW HISTORY (review-history.pdf).** An accounting of the reviewer comments and feedback.

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
