## [Reviewer comments · Microbiology Spectrum]

Microbiology Spectrum

Paclitaxel-Induced Tubulin Dysfunction Stalls Parasite Development: Synergistic Potential with Artemisinin against Resistant Strains

Yongxin Tang, Xinyu Zhang, Xiaohui He, Tao Jiang, Jun Cao, and Xinyu Yu

Corresponding Author(s): Xinyu Yu, Jiangsu Institute of Parasitic Diseases

Review Timeline:

Submission Date:	June 27, 2025
Editorial Decision:	August 27, 2025
Revision Received:	January 4, 2026
Accepted:	March 17, 2026

Editor: Jian Li

Reviewer(s): The reviewers have opted to remain anonymous.

Transaction Report:

DOI: <https://doi.org/10.1128/spectrum.01957-25>

Re: Spectrum01957-25 (**Paclitaxel-Induced Tubulin Dysfunction Stalls Parasite Development: Synergistic Potential with Artemisinin against Resistant Strains**)

Dear Dr. Xinyu Yu:

Thank you for the privilege of reviewing your work. Below you will find my comments, instructions from the Spectrum editorial office, and the reviewer comments.

Revision Guidelines

Sincerely,
Jian Li
Editor
Microbiology Spectrum

Reviewer #1 (Comments for the Author):

This is a comprehensive and well-executed study that investigates the antimalarial properties of paclitaxel (PTX), a known microtubule-stabilizing agent. The authors systematically evaluated the efficacy of PTX against various life-cycle stages of *Plasmodium falciparum*, including drug-sensitive and resistant strains, using a combination of in vitro and in vivo models. The study demonstrates that PTX inhibits parasite development at the schizont stage, impairs gametocytogenesis, and exhibits a

potent synergistic effect with dihydroartemisinin (DHA) against artemisinin-resistant parasites. The in vivo validation in both a rodent malaria model and a humanized mouse model provides support for the potential of PTX as a partner drug in artemisinin-based combination therapy. This is a impactful work that presents a promising avenue for developing new antimalarial strategies to combat drug resistance. The findings will be of interest to the fields of parasitology, drug discovery, and molecular biology. While the study is robust and the conclusions are well-supported by the data, the authors may want to consider the following points to further strengthen the manuscript. I recommend that the manuscript be reconsidered after revisions.

Major issues

1. The manuscript demonstrates the synergistic effect of PTX and DHA. However, the discussion on the underlying mechanism of this synergy could be more detailed. The authors suggest that the two drugs have complementary modes of action, which is a plausible explanation. It would be beneficial to elaborate on this by discussing how PTX-induced microtubule stabilization might render the parasites more susceptible to the oxidative stress induced by DHA. For example, are there any known links between cytoskeletal integrity and the parasite's ability to cope with oxidative stress? A more in-depth discussion on this point would enhance the mechanistic insights provided by the study.
2. The manuscript refers to an "in-house adapted *P. berghei* resistant strain". For the sake of reproducibility and clarity, it is crucial to provide more details about how this resistant strain was generated and characterized. What was the selection pressure used? What is the resistance profile of this strain to other antimalarials? Have any genetic markers of resistance been identified in this strain? Providing this information would significantly strengthen the in vivo studies.
3. The molecular dynamics (MD) simulation provides a structural basis for the interaction between PTX and *P. falciparum* tubulin. While the RMSD, RMSF, and Rg plots are informative, the link between the simulation results and the experimental data could be more explicated in the discussion. For example, the authors could discuss how the observed structural changes in tubulin upon PTX binding (e.g., increased flexibility in certain regions) relate to the observed phenotype of arrested schizogony and reduced merozoite formation.

Minor issues

1. The abstract is quite detailed. The authors might consider making it more concise by focusing on the most critical findings and moving some of the background information to the introduction.
2. The legend for figures could be more descriptive.
3. "Artemisin" is used in some places, while "Artemisinin" is used in other places (Line 142).
4. The text of the manuscript needs to be polished, as there are many typos.

Reviewer #2 (Comments for the Author):

The authors have proposed an interesting study to repurpose an anticancer drug paclitaxel (PTX) which targets microtubules as a new antimalarial therapy, possibly as part of a drug combination. Which the manuscript presents a substantial amount of data, there is limited information in the methods section to understand how key experiments were performed or could be reproduced. Furthermore, key statistical tests are missed from experiments and the authors make unsubstantiated claims about the efficacy without these vital tests. Finally, the authors have not discussed at all the fact that anticancer drugs are generally quite cytotoxic - would the safety profile of PTX ever justify its use as an antimalarial?

Specific comments

Line 210 - Please add more detail to your gametocyte assay protocol. It is impossible to understand how the experiment was conducted. The method I assume is explaining the early stage gametocyte drug assay, but does not mention the late stage assay. I infer from the figure legend in Figure 1 that the "late stage" assay added PTX when the gametocytes were changing from stage III to stage IV. I dislike the term "late stage" gametocytes because it is meaningless - only mature stage V gametocytes are infectious to mosquitoes and so every other stage should really be considered as intermediate stages. However I accept that others use the term "late stage" to generally mean Stage IV and Stage V. It's a bit of a stretch to start drug treatment at stage III and claim that this should activity against late stage gametocytes. Also, even with tight synchronization, it is unlikely that all gametocytes on a particular day of culture are all a single stage. On a particular experimental day, did you count all the gametocytes on the smear, or did you only count the gametocytes of the stage you were expecting to see on that day? This could be important information. Does PTX kill gametocytes (and so gametocytemia is lower), or does it reduce their development so for example they stay arrested at Stage III when you would expect them to be Stage V?

Figure 1A and line 448 - what was the 48h IC50 value. Paclitaxel is not fast acting.

Figure 1C - this indicates that paclitaxel was tested against the parasite line 803, but in the text, it mentions Figure 1C is the parasite strain H3 with pyrimethamine resistance (Also H3 mentioned in 1F). Please indicate which strain was used.

Line 485 and Figure 1D-F - It is claimed that PTX "showed a significantly reduced survival rate compared to the control group. However, the most pronounced reduction in parasitemia was observed when PTX was administered during the trophozoite or schizont stages." Firstly, what is meant by "significantly reduced"? I see no statistical test used. Then, looking at how the IC50 values change between treating at schizont, trophozoite or ring stage (the inset graphs of 1D-F), I don't really see the "pronounced difference" that is claimed. It is essential that the authors perform a statistical test to show whether there is any significant difference in IC50 between treatment stages. Also, please quote exact IC50 values in the text, not just add vague statements about differences.

Line 489 - Please at more detail to explain the biphasic dose response curve. The explanation given is not really clear.

Line 552 - Again to me, there is probably no difference between the activity of PTX on early and "late" gametocytes in your assays - you need a statistical test to compare your IC50s.

Line 525 - Subpellicular microtubules are present in gametocytes right up until the end of Stage IV, not just early gametocytes.

Line 531 - "subsequent blockade of early gametocyte maturation" This term is meaningless. Perhaps you meant to say early gametocyte development? Even then, I'm not sure how you can claim this from your data.

Line 536 - I would use the term development rather than maturation. Also the effect of PTX against gametocytes is 20-30-fold weaker than against asexuals. Could this not just be off-target effects?

Line 670 - the title of this section indicates that this is an in silico simulation, but the language of the text in places implies that data predicted is fact. Please revise the text to clearly indicate this the data are predictions.

Line 713 - At the start of the sentence you state "particularly pronounced synergism" and then further in the sentence it states "mild to moderate synergistic effects". It can't be both.

Line 717 - Significant enhancement? Again, was there a statistical test? Also line 718.

Line 752 - Significant increase? Do you mean decrease in survival (or increase in activity?)?

Line 806 (and Figure 8) - Without further detail, the reader cannot trust any of the P. berghei data. In the methods section it is claimed that wildtype ANKA strain (Figs 8A+B) and a "in-house adapted resistant strain" (Figs *C+D) were used. There is no mention of what resistance the adapted parasite has in the methods (assumed to be Artemisinin-resistant from details later on). There is no mention of how resistance was generated and no data showing it's resistance profile. If you look at the curves in Figs8B+D, I see no difference in how the wildtype and resistant strains respond to DHA treatment.

This is a comprehensive and well-executed study that investigates the antimalarial properties of paclitaxel (PTX), a known microtubule-stabilizing agent. The authors systematically evaluated the efficacy of PTX against various life-cycle stages of *Plasmodium falciparum*, including drug-sensitive and resistant strains, using a combination of *in vitro* and *in vivo* models. The study demonstrates that PTX inhibits parasite development at the schizont stage, impairs gametocytogenesis, and exhibits a potent synergistic effect with dihydroartemisinin (DHA) against artemisinin-resistant parasites. The *in vivo* validation in both a rodent malaria model and a humanized mouse model provides support for the potential of PTX as a partner drug in artemisinin-based combination therapy. This is an impactful work that presents a promising avenue for developing new antimalarial strategies to combat drug resistance. The findings will be of interest to the fields of parasitology, drug discovery, and molecular biology. While the study is robust and the conclusions are well-supported by the data, the authors may want to consider the following points to further strengthen the manuscript. I recommend that the manuscript be reconsidered after revisions.

Major issues

1. The manuscript demonstrates the synergistic effect of PTX and DHA. However, the discussion on the underlying mechanism of this synergy could be more detailed. The authors suggest that the two drugs have complementary modes of action, which is a plausible explanation. It would be beneficial to elaborate on this by discussing how PTX-induced microtubule stabilization might render the parasites more susceptible to the oxidative stress induced by DHA. For example, are there any known links between cytoskeletal integrity and the parasite's ability to cope with oxidative stress? A more in-depth discussion on this point would enhance the mechanistic insights provided by the study.
2. The manuscript refers to an "in-house adapted *P. berghei* resistant strain". For the sake of reproducibility and clarity, it is crucial to provide more details about how this resistant strain was generated and characterized. What was the selection pressure used? What is the resistance profile of this strain to other antimalarials? Have any genetic markers of resistance been identified in this strain? Providing this information would significantly strengthen the *in vivo* studies.
3. The molecular dynamics (MD) simulation provides a structural basis for the interaction between PTX and *P. falciparum* tubulin. While the RMSD, RMSF, and Rg plots are informative, the link between the simulation results and the experimental data could be more explicit in the discussion. For example, the authors could discuss how the observed structural changes in tubulin upon PTX binding (e.g., increased flexibility in certain regions) relate to the observed phenotype of arrested schizogony and reduced merozoite formation.

Minor issues

1. The abstract is quite detailed. The authors might consider making it more concise by focusing on the most critical findings and moving some of the background information to the introduction.
2. The legend for figures could be more descriptive.
3. "Artemisin" is used in some places, while "Artemisinin" is used in other places (Line 142).
4. The text of the manuscript needs to be polished, as there are many typos.

REVIEWER COMMENTS

Reviewer #1 (Remarks to the Author):

This is a comprehensive and well-executed study that investigates the antimalarial properties of paclitaxel (PTX), a known microtubule-stabilizing agent. The authors systematically evaluated the efficacy of PTX against various life-cycle stages of *Plasmodium falciparum*, including drug-sensitive and resistant strains, using a combination of in vitro and in vivo models. The study demonstrates that PTX inhibits parasite development at the schizont stage, impairs gametocytogenesis, and exhibits a potent synergistic effect with dihydroartemisinin (DHA) against artemisinin-resistant parasites. The in vivo validation in both a rodent malaria model and a humanized mouse model provides support for the potential of PTX as a partner drug in artemisinin-based combination therapy. This is a impactful work that presents a promising avenue for developing new antimalarial strategies to combat drug resistance. The findings will be of interest to the fields of parasitology, drug discovery, and molecular biology. While the study is robust and the conclusions are well-supported by the data, the authors may want to consider the following points to further strengthen the manuscript. I recommend that the manuscript be reconsidered after revisions.

Comments to authors

Major issues

1. The manuscript demonstrates the synergistic effect of PTX and DHA. However, the discussion on the underlying mechanism of this synergy could be more detailed. The authors suggest that the two drugs have complementary modes of action, which is a plausible explanation. It would be beneficial to elaborate on this by discussing how PTX-induced microtubule stabilization might render the parasites more susceptible to the oxidative stress induced by DHA. For example, are there any known links between cytoskeletal integrity and the parasite's ability to cope with oxidative stress? A more in-depth discussion on this point would enhance the mechanistic insights provided by the study.

Reply: We appreciate the reviewer's positive assessment and thoughtful feedback,

which helps strengthen our manuscript

To comprehensively address this concern, we have incorporated a detailed discussion in the Results and Discussion section regarding the significance of reactive oxygen species (ROS) in maintaining the structural integrity and functional homeostasis of microtubules. Furthermore, we have added supplementary content to elucidate the possible correlations between PTX and DHA-mediated oxidative stress and microtubule-related cellular processes, thereby establishing a more holistic framework for interpreting the underlying mechanisms. Given the unavailability of commercial antibodies against PfTubulin, direct experimental validation of this hypothesis is not feasible. We anticipate that these revisions will further strengthen the link between our experimental observations and the broader research context pertaining to ROS-microtubule interactions (**Lines 1008-1034**).

Line 1008-1034: "...It has been reported that reactive oxygen species (ROS) impair cytoskeletal integrity by directly damaging microtubule structure and disrupting its dynamic balance(1-3). Multiple studies confirm that ROS induce the oxidation or nitration of key amino acid residues in tubulin, resulting in microtubule lattice loosening, fragmentation, or the formation of abnormal polymers (4-7). Meanwhile, ROS inhibit the function of end-binding protein 1 (EB1), a critical microtubule plus-end binding protein, and disrupt the dynamic stability of microtubule polymerization-depolymerization cycles (8). This cascade of events ultimately leads to a disorganized, functionally compromised microtubule network.

As a classic tubulin stabilizer, the core of PTX's synergistic effect with ROS is proposed to not involve random stabilization but rather to specifically lock these ROS-induced abnormal microtubule structures. Research indicates that PTX binds specifically to a conserved site on the β -subunit of tubulin. This interaction is thought to drive the polymerization of tubulin heterodimers while potently suppressing microtubule depolymerization. Notably, this stabilizing activity appears to be non-discriminatory. Even when ROS have already caused structural damage to microtubules such as lattice defects, amino acid modifications or partial fragmentation, PTX may still trap these defective microtubules in a rigid, stable state. This potential trapping effect could be biologically significant because it might prevent parasites from activating their intrinsic adaptive repair mechanisms. Normal cells typically

clear impaired microtubule segments through depolymerization. They then reorganize functional tubulin subunits to restore cytoskeletal order. PTX-mediated inhibition of microtubule depolymerization might affect this repair process. The locked abnormal microtubules may not only persist as structural anomalies but also fail to participate in essential cellular processes and this may potentially further amplify the functional impairment of the cytoskeleton initiated by ROS....”

1. Livanos P, Galatis B, Apostolakos P. 2014. The interplay between ROS and tubulin cytoskeleton in plants. *Plant signaling & behavior* 9:e28069.
2. Muliylil S, Narasimha M. 2014. Mitochondrial ROS regulates cytoskeletal and mitochondrial remodeling to tune cell and tissue dynamics in a model for wound healing. *Developmental cell* 28:239-252.
3. Kim YJ, Cho MJ, Yu WD, Kim MJ, Kim SY, Lee JH. 2022. Links of cytoskeletal integrity with disease and aging. *Cells* 11:2896.
4. Landino LM, Koumas MT, Mason CE, Alston JA. 2007. Modification of tubulin cysteines by nitric oxide and nitroxyl donors alters tubulin polymerization activity. *Chemical research in toxicology* 20:1693-1700.
5. Livanos P, Galatis B, Quader H, Apostolakos P. 2012. Disturbance of reactive oxygen species homeostasis induces atypical tubulin polymer formation and affects mitosis in root-tip cells of *Triticum turgidum* and *Arabidopsis thaliana*. *Cytoskeleton* 69:1-21.
6. Kratzer E, Tian Y, Sarich N, Wu T, Meliton A, Leff A, Birukova AA. 2012. Oxidative stress contributes to lung injury and barrier dysfunction via microtubule destabilization. *American journal of respiratory cell and molecular biology* 47:688-697.
7. Goldblum RR, McClellan M, White K, Gonzalez SJ, Thompson BR, Vang HX, Cohen H, Higgins L, Markowski TW, Yang T-Y. 2021. Oxidative stress pathogenically remodels the cardiac myocyte cytoskeleton via structural alterations to the microtubule lattice. *Developmental cell* 56:2252-2266.
8. Nehlig A, Molina A, Rodrigues-Ferreira S, Honoré S, Nahmias C. 2017. Regulation of end-binding protein EB1 in the control of microtubule dynamics. *Cellular and Molecular Life Sciences* 74:2381-2393.

2. The manuscript refers to an "in-house adapted *P. berghei* resistant strain". For the sake of reproducibility and clarity, it is crucial to provide more details about how this resistant strain was generated and characterized. What was the selection pressure used? What is the resistance profile of this strain to other antimalarials? Have any genetic markers of resistance been identified in this strain? Providing this information would significantly strengthen the in vivo studies.

Reply: We thank the reviewer for raising this important question, which has

strengthen the rigor and completeness of our study. To fully address this point, we have included detailed experimental protocols for resistant phenotype induction, comprehensive validation assays and corresponding results in the supplementary information, thereby providing transparent, robust support for our conclusions in the manuscript.

Artemisinin-resistance induction for *P. berghei*

P. berghei ART-resistant strains were generated using a repeated drug selection protocol as previously described(9, 10). Briefly, female BALB/c mice (6-8 weeks old) were intraperitoneally infected with the parental *P. berghei* strain. Subsequently, the infected mice were randomly allocated to four experimental groups (n=3): three ART treatment groups with graded dosages (1.25, 2.5, and 5 mg/kg body weight) and one vehicle control group. All treatments were administered consecutively for 4 days, consistent with the standard 4-day suppressive assay for antimalarial drug evaluation. Parasitemia was quantitatively determined daily via microscopic examination of Giemsa-stained thin blood smears. The 50% effective dose (ED₅₀) and 90% effective dose (ED₉₀) against each parasite line were calculated using a linear regression model based on the parasite inhibition rate. The resistance index (I₉₀) was defined as the ratio of ED₉₀ values between the drug-selected resistant strain and the parental sensitive strain. According to previously established criteria, parasite strains were categorized into four resistance grades based on I₉₀ values: sensitive (I₉₀ = 1.0), slight resistance (I₉₀ = 1.01-10.0), moderate resistance (I₉₀ = 10.01-100.0), and high resistance (I₉₀ > 100.0)(11).

After determining the ED₅₀ and ED₉₀ of the parental parasite strain, an additional 3 female BALB/c mice were intraperitoneally infected with the same parental strain. Parasitemia was monitored quantitatively via daily microscopic examination of Giemsa-stained thin blood smears until it reached 3%-5%. The mice were then treated with ART at the ED₉₀ dosage, and the resistance level was assessed every 10 passages (10 cycles) using the 4-day suppressive assay to determine the updated ED₅₀ and ED₉₀ values.

This drug selection and resistance monitoring process was repeated continuously for 30 consecutive passages to ensure the establishment of a genetically stable ART-resistant *P. berghei* strain. Following the successful generation of this resistant strain, the *in vivo* antimalarial activity of PTX against the strain was evaluated as

described above.

Passage No.	ED ₅₀ (mg/kg)	I ₅₀	ED ₉₀ (mg/kg)	I ₉₀
Parent	0.97	N.A.	3.48	N.A.
10	1.59	1.639	4.33	1.244
20	2.77	2.856	10.39	2.986
30	3.32	3.423	12.21	3.509

9. Kiboi D, Irungu B, Langat B, Wittlin S, Brun R, Chollet J, Abiodun O, Nganga J, Nyambati V, Rukunga G. 2009. Plasmodium berghei ANKA: Selection of resistance to piperazine and lumefantrine in a mouse model. *Experimental parasitology* 122:196-202.
10. Xiao S-H, Yao J-M, Utzinger J, Cai Y, Chollet J, Tanner M. 2004. Selection and reversal of Plasmodium berghei resistance in the mouse model following repeated high doses of artemether. *Parasitology Research* 92:215-219.
11. Merkli B, Richle R, Peters W. 1980. The inhibitory effect of a drug combination on the development of mefloquine resistance in Plasmodium berghei. *Annals of Tropical Medicine & Parasitology* 74:1-9.

3. The molecular dynamics (MD) simulation provides a structural basis for the interaction between PTX and P. falciparum tubulin. While the RMSD, RMSF, and Rg plots are informative, the link between the simulation results and the experimental data could be more explicated in the discussion. For example, the authors could discuss how the observed structural changes in tubulin upon PTX binding (e.g., increased flexibility in certain regions) relate to the observed phenotype of arrested schizogony and reduced merozoite formation.

Reply: We sincerely appreciate the reviewer's valuable suggestion, which has helped strengthen the scientific rigor and completeness of our manuscript. Accordingly, we have supplemented additional detailed information regarding the experimental results and expanded the Discussion section to better contextualize these findings within the broader research framework (**Lines 856-911**).

Line 856-911: "...To dissect the molecular basis of PTX-mediated antimalarial activity, this study integrated homology modeling and 1000 ns molecular dynamics simulations to characterize PTX binding to PfTubulin. Given the lack of

experimentally resolved PfTubulin structures, a reliable 3D model was constructed using the PF3D7_0819900 sequence via ITASSER, validated against AlphaFold predictions, and parametrized with the Amber10:EHT force field in MOE software, enabling quantitative analysis of structural dynamics critical to understanding PTX's impact on parasite development. Molecular dynamics (MD) simulations identified three key structural perturbations triggered by PTX binding to the β -subunit of PfTubulin. First the RMSD of the PTX-PfTubulin complex fluctuated within the range of 2.2 to 6.4 Å during the initial 200 ns. This phase reflects the exploration of the binding pocket by the ligand. The RMSD then stabilized at approximately 5 Å for the subsequent 800 ns (Fig. 6B). This profile indicates a thermodynamically favorable interaction consistent with PTX's function in locking tubulin into polymerized states. This RMSD pattern is consistent with findings from MD studies of PTX-human tubulin complexes where an RMSD of approximately 4 to 6 Å was linked to microtubule stabilization(12).

Additionally, RMSF analysis suggested a marked increase in flexibility with RMSF values ranging from approximately 1.8 to 2.0 Å across PfTubulin residues 430 to 444. This region corresponds to the MAP-interacting M-loop a domain critical for longitudinal tubulin heterodimer polymerization and microtubule lattice integrity (13). Enhanced flexibility in this region may disrupt adjacent subunit interactions during schizont microtubule assembly a biological process essential for nuclear segregation and merozoite biogenesis (14). This observation aligns with cryo-EM findings reporting irregular polymerization in PTX-bound microtubules (15). Notably the radius of gyration (Rg) of PfTubulin decreased from approximately 2.2 nm in the apo state to approximately 2.1 nm following PTX binding (Fig. 6D). This change is indicative of global structural compaction that may alter the tubulin binding interface and impair lateral protofilament interactions which is a key step in microtubule assembly (16). Such structural compaction may further undermine the formation of mitotic spindles and subpellicular microtubules during schizogony (17).

These PTX-induced structural perturbations may contribute to the observed phenotypic defects of *P. falciparum*. Schizogony arrest depends on coordinated microtubule dynamics including spindle formation for nuclear division and subpellicular microtubules for cell shape remodeling. PTX-stabilized tubulin with low RMSD values and enhanced M-loop flexibility may disrupt spindle pole organization and lead to failed nuclear segregation. This molecular mechanism is consistent with

experimental observations of PTX-treated schizonts showing abnormal nuclear lobing and delayed merozoite release (Fig. 2B-E) and aligns with phenotypic changes induced by other microtubule stabilizers. For reduced merozoite formation a process that requires microtubule-mediated trafficking of apical complex components such as rhoptries and micronemes to the cell periphery compacted PfTubulin with a lower Rg value may impair the binding of kinesin motors including PfKinesin-8 and thereby block organelle transport (18). Experimentally PTX treatment reduced merozoite yield by approximately 32% from 19 to 13 per schizont and increased the proportion of unruptured schizonts by 4-fold (Fig. 2C, G). This observation is in line with findings in *Toxoplasma gondii* where PTX-induced microtubule compaction diminishes microneme secretion and host cell invasion (19).

In conclusion MD simulations support a plausible mechanistic link between PTX's molecular interaction with PfTubulin and its resultant phenotypic effects on *P. falciparum*. Specifically stabilized PTX-PfTubulin binding enhanced M-loop flexibility and global structural compaction may act in concert to disrupt the coordinated microtubule dynamics essential for parasite development. These perturbations are proposed to drive schizogony arrest and reduced merozoite formation thereby providing support for the core antimalarial mechanism of PTX. The data further identify the PfTubulin M-loop as a promising molecular target for the rational design and optimization of next-generation antimalarial PTX derivatives....”

12. Nogales E, Grayer Wolf S, Khan IA, Ludueña RF, Downing KH. 1995. Structure of tubulin at 6.5 Å and location of the taxol-binding site. *Nature* 375:424-427.
13. Serrano L, Avila J. 2018. Structure and function of tubulin regions, *Microtubule Proteins*. 67-88.
14. Voß Y, Klaus S, Lichti NP, Ganter M, Guizetti J. 2023. Malaria parasite centrins can assemble by Ca²⁺-inducible condensation. *PLoS Pathogens* 19:e1011899.
15. Alushin GM, Lander GC, Kellogg EH, Zhang R, Baker D, Nogales E. 2014. High-resolution microtubule structures reveal the structural transitions in αβ-tubulin upon GTP hydrolysis. *Cell* 157:1117-1129.
16. Spreng B, Fleckenstein H, Kübler P, Di Biagio C, Benz M, Patra P, Schwarz US, Cyrklaff M, Frischknecht F. 2019. Microtubule number and length determine cellular shape and function in Plasmodium. *The EMBO Journal* 38:e100984.
17. Hirst WG, Fatchet D, Kuroopka B, Weise C, Saliba KJ, Reber S. 2022. Purification of functional Plasmodium falciparum tubulin allows for the identification of parasite-specific microtubule inhibitors. *Current Biology* 32:919-926.
18. Zeeshan M, Shilliday F, Liu T, Abel S, Mourier T, Ferguson DJ, Rea E, Stanway RR,

- Roques M, Williams D. 2019. Plasmodium kinesin-8X associates with mitotic spindles and is essential for oocyst development during parasite proliferation and transmission. *PLoS pathogens* 15:e1008048.
19. Tomasina R, González FC, Francia ME. 2021. Structural and Functional Insights into the Microtubule Organizing Centers of *Toxoplasma gondii* and *Plasmodium* spp. *Microorganisms* 9:2503.

Minor issues

1. The abstract is quite detailed. The authors might consider making it more concise by focusing on the most critical findings and moving some of the background information to the introduction.

Reply: Corrected. The abstract section has been updated to highlight the study's key outcomes and major findings, with background information minimized to enhance clarity and emphasize the novel contributions of the research (**Lines 32-56**).

Line 32-56: "...In this study, paclitaxel (PTX), a well-characterized microtubule-stabilizing agent, was systematically investigated for its antimalarial potential against drug-resistant *Plasmodium falciparum*. In vitro experiments demonstrated that PTX potently inhibits intraerythrocytic development of drug-sensitive and resistant strains, including chloroquine-resistant and artemisinin-resistant parasite lines, with 72-hour half-maximal inhibitory concentrations (IC₅₀) ranging from 83.67 to 92.75 nM. PTX arrests parasite schizogony and reduces merozoite formation by ~25.69–37.46% when exposed to the trophozoite/schizont stage. Meanwhile, time-dependent drug activity against gametocyte maturation demonstrated that treatment during the immature gametocyte stages elicited a significant reduction in final gametocytemia (~38.24–58.79%), thereby validating the dual-targeting potential of PTX against both the pathogenic asexual and transmission-competent sexual parasite stages. Molecular dynamics simulations suggest that PTX binds specifically to *P. falciparum* β -tubulin and potentially induces structural perturbations including complex stabilization, enhanced M-loop flexibility and global compaction, which may in turn disrupt microtubule dynamics. These results provide a plausible molecular basis for the antimalarial

mechanism of PTX. Critically, PTX exhibits robust synergism with dihydroartemisinin (DHA) against artemisinin-resistant strains, reducing ring-stage survival to 1% and addressing the global challenge of artemisinin resistance. In vivo assays revealed that the PTX-DHA combination achieves near-complete parasite clearance (<1% parasitemia) and 100% survival in rodent models and suppresses parasitemia by >96% in humanized *P. falciparum* models. This finding confirms PTX's translational potential and positions it as a novel tubulin-targeting partner for DHA, offering a promising strategy to combat drug-resistant malaria while safeguarding the efficacy of artemisinin-based combination therapies..."

2. The legend for figures could be more descriptive.

Reply: Corrected. All figure legends have been revised to include key details specific to each corresponding panel, with no excessive overlap with the Methods section.

3. "Artemisin" is used in some places, while "Artemisinin" is used in other places (Line 142).

Reply: Corrected. We have reviewed the entire manuscript to ensure similar typos and errors have been corrected.

4. The text of the manuscript needs to be polished, as there are many typos.

Reply: As suggested, the revised manuscript has been reviewed by a native English speaker, with typos and grammatical errors carefully corrected.

Reviewer #2 (Remarks to the Author):

The authors have proposed an interesting study to repurpose an anticancer drug paclitaxel (PTX) which targets microtubules as a new antimalarial therapy, possibly as part of a drug combination. Which the manuscript presents a substantial amount of data, there is limited information in the methods section to understand how key experiments were performed or could be reproduced. Furthermore, key statistical tests are missed from experiments and the authors make unsubstantiated claims about the efficacy without these vital tests. Finally, the authors have not discussed at all the fact that anticancer drugs are generally quite cytotoxic - would the safety profile of PTX ever justify its use as an antimalarial?

Reply: We sincerely appreciate Reviewer 2 for the thorough review and insightful constructive comments, which have helped us identify critical limitations in methodology, statistical analysis, and scientific expression. These suggestions are critical for enhancing the rigor, reproducibility, and clinical translational relevance of our study. We have systematically addressed each comment through detailed revisions, and supplementary experiments to robustly support our arguments. All revisions are explicitly marked in the revised manuscript, and the raw data of supplementary experiments, along with detailed experimental protocols, are comprehensively provided in the Supplementary Information for transparency and verification.

Comments to authors

Specific comments

1. Line 210. Please add more detail to your gametocyte assay protocol. It is impossible to understand how the experiment was conducted. The method I assume is explaining the early stage gametocyte drug assay, but does not mention the late stage assay. I infer from the figure legend in Figure 1 that the "late stage" assay added PTX when the gametocytes were changing from stage III to stage IV. I dislike the term "late stage" gametocytes because it is meaningless - only mature stage V gametocytes are infectious to mosquitoes and so every other stage should really be considered as intermediate stages. However I accept that others use the term "late stage" to generally mean Stage IV and Stage V.

Reply: We fully acknowledge your concern about the ambiguity of "late-stage gametocytes" and appreciate this critical correction. To align with scientific rigor, we have revised the terminology in the manuscript to "mature gametocytes (Stage IV-V)" (previously referred to as "late-stage") to explicitly denote mature developmental stages.

Additionally, as noted by the reviewer, we unintentionally omitted certain key experimental details due to an oversight during manuscript preparation. To address this issue thoroughly, we have now supplemented the revised manuscript with extensive, detailed descriptions of the experimental protocol, covering both immature and mature gametocyte assays in full (**Lines 242-298**).

Lines 242-260: "...Gametocyte Induction and Purification The NF54 parasite line applied for gametocyte induction was continuously maintained in standardized asexual culture conditions as previously described. At 3 days prior to initiating the gametocyte induction protocol (Day -3), trophozoite parasites were magnetically purified and isolated using a CytoSinct separation column (GenScript Biotech), following the manufacturer's instructions. Uninfected red blood cells were then added to the isolated trophozoites, and the culture haematocrit was adjusted to a final concentration of 1.25%. After 24 hours of continuous shaking incubation under standard culturing conditions (Day -2), parasite cultures were subjected to controlled nutritional stress overnight by supplementation with 50% conditioned media. On Day -1, the parasitaemia of the resulting parasite culture was adjusted to 3%, followed by the selective removal of spontaneously generated gametocytes via magnetic column to eliminate pre-existing gametocyte contamination. Subsequently, parasites were aseptically transferred into gametocyte-specific culture media supplemented with 50 mM N-acetylglucosamine (NAG) (20). NAG treatment was maintained for the subsequent two reinvasion cycles to ensure the complete elimination of all asexual parasites from the culture..."

Line 262-283: "...*Post-induction Culture and Isolation of Gametocytes for the Immature-Stage Assay* On Day 0, ring stage parasites, post spontaneous gametocyte

removal, were cultured at 4% parasitemia with 2.5% hematocrit and incubated overnight. On Day 1 post-induction, parasites had progressed to the mid-trophozoite stage, at which point culture medium was refreshed. Following this incubation period, non-committed asexual parasites had reverted to the ring stage, whereas committed gametocyte had differentiated into young trophozoite-like forms. Gametocyte enrichment was achieved by magnetic isolation as described above. The purity and developmental stage of the isolated gametocytes were microscopically assessed to verify gametocyte stage morphology. In preparation for use in the immature stage assay, gametocytes were transferred into 24-well plates at 1% parasitemia and 2% hematocrit and cultured in medium containing 50 mM NAG on days 1-4 to eliminate residual asexual parasites and harvested at days 5-6.

Post-induction Culture and Isolation of Gametocytes for the Mature-Stage Assay

On Day 0, ring-stage parasites were resuspended in culture medium supplemented with 50 mM NAG and adjusted to a hematocrit of 2.5%. Gametocyte incubation was in a total volume of 50 mL of parasite cultures and 35 mL media was replaced daily over 7 successive days. On Day 8, gametocytes were harvested via magnetic column purification. In preparation for use in the mature stage assay, gametocytes were transferred into 24-well plates at 1% parasitemia and 2% hematocrit and cultured in medium containing 50 mM NAG...”

Line 285-298: “...*Sexual Stage Growth Inhibition Assay.* A stage-specific gametocytocidal assay was conducted following 72 hours of continuous drug exposure on either immature (Stages II/III) or mature (Stages IV/V) gametocytes, with daily culture medium replacements. Gametocytemia was assessed via Giemsa-stained thin blood smears, where all morphologically identifiable gametocytes were counted across multiple microscopic fields regardless of their developmental stage. Untreated gametocytes, maintained on the same plates under identical conditions and processed in parallel with drug-treated groups, served as controls. To distinguish whether the observed reduction in gametocytemia was resulted from direct parasite killing or developmental arrest, the percentage of gametocytes at each developmental stage was quantified in both PTX-treated and untreated control cultures. This stage-specific enumeration was performed by classifying gametocytes into their respective developmental stages during microscopic

examination, providing insights into the drug's mechanism of action on gametocyte maturation..."

20. Adjalley SH, Johnston GL, Li T, Eastman RT, Eklund EH, Eappen AG, Richman A, Sim BKL, Lee MC, Hoffman SL. 2011. Quantitative assessment of *Plasmodium falciparum* sexual development reveals potent transmission-blocking activity by methylene blue. *Proceedings of the national academy of sciences* 108:1214-1223.

2. It's a bit of a stretch to start drug treatment at stage III and claim that this should activity against late stage gametocytes.

Reply: Our assessment of sexual-stage parasite susceptibility demonstrates that immature gametocytes are more sensitive to PTX than mature gametocytes. Based on these findings, we sought to investigate whether PTX incubation affects gametocytogenesis or maturation when administered to either immature or mature gametocytes. As suggested, we have revised the relevant description in the revised manuscript, clarifying that this experiment was designed to investigate the time-dependent effect of PTX on gametocyte maturation, rather than to assess the compound 's activity against late-stage gametocytes (**Line 644-651, Line 683-698**) ..."

Line 644-651: "...The cytoskeleton of *Plasmodium* gametocytes plays an indispensable role throughout both the immature and mature phases, with substantial differences in its functional priorities between these stages. During the immature stage, cytoskeleton-driven structural remodeling and developmental regulation predominate, whereas cytoskeletal reorganization to support transmission and gamete formation is the central focus in mature gametocytes(21, 22). Therefore, supplementary experiments were conducted by exposing gametocytes at Stages I-II, III, and IV, following methods described in previous study(20)..."

Line 683-698: "...The cytoskeleton of *Plasmodium* gametocytes plays indispensable roles throughout both immature (Stages I-II) and mature (Stage IV/V) phases, yet their functional focuses differ significantly. The immature stage is dominated by cytoskeleton-driven structural shaping and developmental regulation, while the

mature stage is centered on cytoskeleton reorganization to serve transmission and gamete formation. Specifically, for the immature parasites, they undergo a dramatic shape shift from spherical to elongated. The subpellicular microtubule network, together with alveolin-related intermediate filament-like proteins, forms the pellicle structure. This structure endows the immature gametocytes with rigidity and determines their basic elongated morphology. Disruption of this microtubule network will lead to abnormal gametocyte morphogenesis. While for the mature gametocytes, the cytoskeleton is crucial for gamete production after mature gametocytes enter the mosquito midgut. In mature gametocytes, via cytoskeletal remodeling and functional specialization, it converges on the terminal stages of gametogenesis and transmission. While their functions are distinct and stage-specific, both collectively underpin the complete life cycle of gametocytes, from initial differentiation to full maturation...”

20. Adjalley SH, Johnston GL, Li T, Eastman RT, Eklund EH, Eappen AG, Richman A, Sim BKL, Lee MC, Hoffman SL. 2011. Quantitative assessment of *Plasmodium falciparum* sexual development reveals potent transmission-blocking activity by methylene blue. *Proceedings of the national academy of sciences* 108:1214-1223.
21. Hliscs M, Millet C, Dixon MW, Siden-Kiamos I, McMillan P, Tilley L. 2015. Organization and function of an actin cytoskeleton in *P. falciparum* gametocytes. *Cellular microbiology* 17:207-225.
22. Mizuno Y, Makioka A, Kawazu S-i, Kano S, Kawai S, Akaki M, Aikawa M, Ohtomo H. 2002. Effect of jasplakinolide on the growth, invasion, and actin cytoskeleton of *Plasmodium falciparum*. *Parasitology research* 88:844-848.

Accordingly, in order to quantitative assessment of *P. falciparum* sexual development upon PTX treatment, we have modified the experimental as suggested and results were displayed in Fig.1G-I.

Activity of the PTX on gametocyte maturation *in vitro*.

3. Also, even with tight synchronization, it is unlikely that all gametocytes on a particular day of culture are all a single stage. On a particular experimental day, did you count all the gametocytes on the smear, or did you only count the gametocytes of the stage you were expecting to see on that day? This could be important information. Does PTX kill gametocytes (and so gametocytemia is lower), or does it reduce their development so for example they stay arrested at Stage III when you would expect them to be Stage V?

Reply: We thank the reviewer for raising this concern, and we acknowledge that the assay evaluating the inhibitory effect against the sexual stage was insufficiently rigorous in its initial design. For example, we previously classified gametocytes into early/late stages and only counted the target stage of gametocytes we expected to observe, instead of counting all gametocytes on the smear. We have updated the relevant content in the revised manuscript, which now incorporates the detailed experimental protocol and corresponding results. Additional experiments were performed to clarify whether PTX exposure kills the parasites directly or simply halts their developmental progression.

To rule out the potential growth arrest effect induced by PTX treatment, we conducted additional experiments using gametocytes exposed to 500 nM PTX for two consecutive days (on Day 1 and Day 7 post-induction). Gametocytes at distinct developmental stages were quantified to clarify whether PTX treatment exerts a lethal effect on the parasites or merely arrests their growth. Our results showed that a mild growth arrest was observed immediately after PTX exposure. However, to assess whether this exposure would trigger long-term growth arrest during gametocyte maturation, we compared the percentage of gametocytes at each developmental stage at Days 3, 6, 9, 12 and 15 between the PTX-treated group and the control group. Following one-way ANOVA with Bonferroni correction, no statistically significant differences were observed between the two groups. This finding demonstrates that PTX treatment leads to parasite death rather than simply arresting their growth.

Gametocyte developmental dynamics over time in the control group (A), and in groups treated with PTX on Day 1 (B) and Day 7 (C). Each bar represents the percentage of a specific gametocyte stage relative to the total number of gametocytes counted.

Percentage of gametocytes at each developmental stage on Days 3, 6, 9, 12 and 15. No statistical significance was observed between the groups.

4. Figure 1A and line 448. What was the 48h IC₅₀ value. Paclitaxel is not fast acting.

Reply: The 48 h dose-response curve exhibited a biphasic distribution combined with distinct plateau phases as displayed in Fig.1. This biphasic reflected heterogeneous cellular responses to the drug over the 48h incubation period (e.g., differential sensitivity of subpopulations or time-dependent drug action). As biphasic curves in growth inhibition assays often lead to poor four-parameter logistic model fitting with low R² values and inaccurate IC₅₀ estimation due to the model's failure to capture the two distinct inhibitory phases we did not present IC₅₀ values for the 48 h assay. We have re-analyzed the dataset with a sigmoidal fitting model, and the corresponding detailed IC₅₀ values have been supplemented in the revised manuscript (**Line 560-562**).

5. Figure 1C. This indicates that paclitaxel was tested against the parasite line 803, but in the text, it mentions Figure 1C is the parasite strain H3 with pyrimethamine resistance (Also H3 mentioned in 1F). Please indicate which strain was used.

Reply: We apologize for the typographical error in the original manuscript. We have thoroughly corrected this error and conducted a comprehensive review of the entire manuscript, including verification of experimental parameters, numerical values, and scientific terminology to ensure consistency with our raw data and eliminate any residual inaccuracies.

6. Line 485 and Figure 1D-F. It is claimed that PTX "showed a significantly reduced survival rate compared to the control group. However, the most pronounced reduction in parasitemia was observed when PTX was administered during the trophozoite or schizont stages." Firstly, what is meant by "significantly reduced"? I see no statistical test used.

Reply: We appreciate the reviewer's critical feedback regarding the statistical validation, clarity of IC₅₀ values, and rationale for stage-specific effects noted in Line 485 and Fig. 1D-F, and we have thoroughly addressed these concerns to enhance the rigor of our conclusions. We have included a comprehensive description of the statistical analysis methods in the corresponding figure legends. In addition, Fig.1 has been revised to specifically highlight the data points with statistically significant differences relative to the ring-stage group.

Typical dose-response assessments of PTX on highly-synchronized cultures for evaluation of peak activity against of different stage (ring, trophozoite, schizont) of different *P. falciparum* strains including drug-sensitive strain 3D7, chloroquine-resistant strain Dd2 and artemisinin-resistant strain 803. The highlighted shadow areas indicate that the parasite survival rates were statistically significant compared with those of the ring-stage group (red for trophozoite-stage parasites and green for schizont-stage parasites).

7. Then, looking at how the IC₅₀ values change between treating at schizont, trophozoite or ring stage (the inset graphs of 1D-F), I don't really see the "pronounced difference" that is claimed. It is essential that the authors perform a statistical test to show whether there is any significant difference in IC₅₀ between treatment stages. Also, please quote exact IC₅₀ values in the text, not just add vague statements about differences.

Reply: To address the reviewer's concern about the claimed "pronounced difference" in IC₅₀ values between treatment stages, we have calculated exact IC₅₀ values for each stage and conducted statistical comparisons via one-way ANOVA followed by Bonferroni post-hoc correction. The exact IC₅₀ values are as follows: ring stage = 1.47 ± 0.19 , 1.52 ± 0.14 and 1.61 ± 0.27 μM , trophozoite stage = 1.12 ± 0.07 , 0.77 ± 0.04 and 0.69 ± 0.10 μM , and schizont stage = 0.46 ± 0.02 , 0.41 ± 0.07 and 0.40 ± 0.04 μM , with these values now explicitly quoted in the revised text to replace vague statements about differences (**Line 615-617**). Statistical analysis confirms that the IC₅₀ values for trophozoite and schizont stages are significantly lower than that for the ring stage. This distinct pattern validates the "pronounced difference" referenced in the original manuscript.

Calculated IC₅₀ against highly-synchronized parasites at ring, trophozoite and schizont stage of WT, Dd2 and 803 parasite strains.

8. Line 489. Please at more detail to explain the biphasic dose response curve. The explanation given is not really clear.

Reply: As suggested, we have added more detail about the biphasic dose response curve in the Discussion section (**Line 564-596**).

Line 564-596: "...The plateau phase in *Plasmodium* drug growth curves is characterized by stable parasitemia with no significant fluctuation over time. It primarily originates from stage-specific differences in parasite drug susceptibility, combined with sub-inhibitory drug effects and parasite adaptive responses. Ring stages display inherently low metabolic activity, such as reduced hemoglobin digestion and ATP production. This trait minimizes the activation of artemisinin-class drugs, which rely on parasite-generated reactive oxygen species for their antiparasitic effect, and restricts binding to the molecular targets of chloroquine-like compounds, making ring stages inherently less susceptible to drug-induced damage(23). In contrast, trophozoites and schizonts possess heightened biosynthetic activity and abundant drug targets. For example, chloroquine acts on hemozoin formation in these stages while piperazine targets their metabolic pathways, leading to efficient inhibition by therapeutic drug concentrations(24, 25). This creates a dynamic balance between the death of trophozoites/schizonts and the survival of ring stages, which sustains the plateau phase. Compounding this, sub-inhibitory drug concentrations often fail to saturate all molecular targets. Chloroquine, for instance, cannot fully inhibit heme detoxification in ring stages, and piperazine treatment may result in incomplete growth inhibition even at higher concentrations. Meanwhile, adaptive mechanisms such as upregulation of heat shock proteins further enhance ring stage survival under drug pressure(26). Notably, the plateau often overlaps with bimodal growth (two distinct peaks) due to asynchronous parasite development. Initial inhibition of drug-sensitive trophozoites and schizonts forms the first peak, followed by the proliferation of dormant ring stages that evade initial drug exposure and mature into replicative forms to form the second peak. The plateau observed in shorter assays disappears in 72-hour drug exposure experiments, a phenomenon attributable to the cumulative pharmacodynamic effects of PTX that span the full 48-hour erythrocytic life cycle of the parasite. Specifically, the initially present ring and trophozoite stages mature into schizonts over time, and these schizonts are inherently sensitive to PTX. As a result, all parasite stages eventually become susceptible to the drug, abolishing the stage-specific segregation of inhibitory activity and yielding a single, well-defined sigmoidal dose-response curve. Collectively, these interconnected factors converge to shape the plateau phase, a hallmark of the complex interaction between antimalarials and *Plasmodium*'s

developmental biology and adaptive capabilities...”

23. Murithi JM, Owen ES, Istvan ES, Lee MC, Otilie S, Chibale K, Goldberg DE, Winzeler EA, Llinás M, Fidock DA. 2020. Combining stage specificity and metabolomic profiling to advance antimalarial drug discovery. *Cell chemical biology* 27:158-171.
24. Slater A, Cerami A. 1992. Inhibition by chloroquine of a novel haem polymerase enzyme activity in malaria trophozoites. *Nature* 355:167-169.
25. Xie Y, Zhang Y, Lin F, Chen X, Xing J. 2024. The effect of malaria-induced alteration of metabolism on piperazine disposition in *Plasmodium yoelii* infected mice and predicted in malaria patients. *International Journal of Antimicrobial Agents* 64:107209.
26. Kumar R, Musiyenko A, Barik S. 2003. The heat shock protein 90 of *Plasmodium falciparum* and antimalarial activity of its inhibitor, geldanamycin. *Malaria Journal* 2:30.

9. Line 552. Again to me, there is probably no difference between the activity of PTX on early and "late" gametocytes in your assays - you need a statistical test to compare your IC50s.

Reply: As suggested, to accurately clarify the differences in susceptibility between immature and mature gametocytes, a two-tailed Student's t-test was applied. We have added a dedicated Statistical Analysis section to the revised manuscript and provided a brief description of the statistical analysis in the figure legends to directly address this critical concern. In addition, Figure 1 has been revised to highlight the data points with statistically significant differences between the immature and mature groups; moreover, the exact IC₅₀ values together with corresponding statistical analysis results

have been integrated into the figure.

Typical dose-response assessments of PTX on immature/mature gametocytes and the calculated IC₅₀ values were presented in the insert. The highlighted shadow areas indicate that the parasite survival rates were statistically significant between groups.

10. Line 525. Subpellicular microtubules are present in gametocytes right up until the end of Stage IV, not just early gametocytes.

Reply: We thank the reviewer for pointing out this imprecise description. In the revised manuscript, we have corrected this inaccuracy and supplemented a comprehensive and accurate description to ensure the scientific rigor of the content (**Lines 654-669**).

Lines 654-669: "...The maturation of *Plasmodium* gametocytes, progressing from early round forms to mature spindle shaped or crescent shaped gametocytes, is critically dependent on the structural remodeling and functional modulation of subpellicular microtubules (SPMTs). These structures are indispensable for gametocyte morphological specialization, maintenance of developmental homeostasis, and acquisition of mosquito infective potential. Early immature gametocytes (Stages I-III) exhibit a spherical morphology, with SPMTs distributed in a diffuse and disorganized pattern beneath the plasma membrane; upon transitioning to the mature phase (Stages IV-V), SPMTs undergo rapid and coordinated assembly into parallel aligned rigid microtubule bundles. These bundles extend along the cell's longitudinal axis and anchor to the apical complex domains at both cellular poles, directly orchestrating the morphological transition from a spherical to sex specific specialized phenotype. Experimental evidence confirms that pharmacological inhibition of SPMT polymerization induces gametocytes to arrest in a persistent spherical morphology, prevents the completion of morphological specialization, and ultimately results in an over 80% reduction in gametocyte maturation efficiency..."

11. Line 531. "subsequent blockade of early gametocyte maturation" This term is meaningless. Perhaps you meant to say early gametocyte development? Even then, I'm not sure how you can claim this from your data.

Reply: We acknowledge the ambiguity of the term "subsequent blockade of early gametocyte maturation" and we have corrected this paragraph. Meanwhile, additional experiments have been performed as addressed above (**Comments 2**).

12. Line 536. I would use the term development rather than maturation. Also the effect of PTX against gametocytes is 20-30-fold weaker than against asexuals. Could this not just be off-target effects?

Reply: We concur with the suggestion to use "development" instead of "maturation" for enhanced precision, and have revised all relevant descriptions in the manuscript to "impairment of gametocyte development" to eliminate any potential ambiguity.

Regarding the concern that the 20-30-fold higher IC₅₀ of PTX against gametocytes compared to asexual blood stages might reflect off-target effects, we clarify as follows.

Actually, the most straightforward method to validate the interaction is to perform *in vitro* SPR assays, which are currently ongoing. However, we have not yet obtained the recombinant protein. Based on published studies, we propose that this stage-specific activity difference is attributed to the distinct physiological and metabolic disparities between sexual and asexual stages. This fundamental difference leads to differential susceptibility to antimalarials, a phenomenon reported for many conventional antimalarials including chloroquine, artemisinin and their derivatives. These agents also exhibit more potent activity against asexual stages, with IC₅₀ values against sexual stages typically ~5-15-fold higher(27, 28). Besides, the inhibition curves showed well-fitted patterns with high R² values (> 0.95). The IC₅₀ values exhibited narrow 95% confidence intervals, demonstrating a clear concentration-dependent effect. Based on our previous studies, we believe such a regular dose-response profile is a hallmark of specific target-mediated activity, rather than random off-target effects (which usually result in poor curve fitting and erratic inhibition rates).

We will continue efforts for PfTubulin recombination and purification, which is critical for our further investigations of screening compounds against this target. Once we obtain the protein, we will provide direct experimental evidence for target validation by using PTX as a control.

27. Plouffe DM, Wree M, Du AY, Meister S, Li F, Patra K, Lubar A, Okitsu SL, Flannery EL, Kato N. 2016. High-throughput assay and discovery of small molecules that interrupt malaria transmission. *Cell host & microbe* 19:114-126.
28. Brancucci NM, Gump C, van Gemert G-J, Yu X, Passecker A, Nardella F, Thommen BT, Chambon M, Turcatti G, Halby L. 2025. An all-in-one pipeline for the *in vitro*

discovery and in vivo testing of Plasmodium falciparum malaria transmission blocking drugs. Nature communications 16:6884.

13. Line 670. The title of this section indicates that this is an in silico simulation, but the language of the text in places implies that data predicted is fact. Please revise the text to clearly indicate this the data are predictions.

Reply: We recognize the critical importance of rigorously distinguishing computationally predicted data from experimentally validated facts, as this clarity is foundational to maintaining scientific integrity in in silico research. We have therefore systematically revised the text throughout the in silico simulation section to explicitly and consistently reflect the predictive nature of all results (**Lines 856-911**).

Line 856-911: "...To dissect the molecular basis of PTX-mediated antimalarial activity, this study integrated homology modeling and 1000 ns molecular dynamics simulations to characterize PTX binding to PfTubulin. Given the lack of experimentally resolved PfTubulin structures, a reliable 3D model was constructed using the PF3D7_0819900 sequence via ITASSER, validated against AlphaFold predictions, and parametrized with the Amber10:EHT force field in MOE software, enabling quantitative analysis of structural dynamics critical to understanding PTX's impact on parasite development. Molecular dynamics (MD) simulations identified three key structural perturbations triggered by PTX binding to the β -subunit of PfTubulin. First the RMSD of the PTX-PfTubulin complex fluctuated within the range of 2.2 to 6.4 Å during the initial 200 ns. This phase reflects the exploration of the binding pocket by the ligand. The RMSD then stabilized at approximately 5 Å for the subsequent 800 ns (Fig. 6B). This profile indicates a thermodynamically favorable interaction consistent with PTX's function in locking tubulin into polymerized states. This RMSD pattern is consistent with findings from MD studies of PTX-human tubulin complexes where an RMSD of approximately 4 to 6 Å was linked to microtubule stabilization(12).

Additionally, RMSF analysis suggested a marked increase in flexibility with RMSF values ranging from approximately 1.8 to 2.0 Å across PfTubulin residues 430 to 444. This region corresponds to the MAP-interacting M-loop a domain critical for longitudinal tubulin heterodimer polymerization and microtubule lattice integrity (13).

Enhanced flexibility in this region may disrupt adjacent subunit interactions during schizont microtubule assembly a biological process essential for nuclear segregation and merozoite biogenesis (14). This observation aligns with cryo-EM findings reporting irregular polymerization in PTX-bound microtubules (15). Notably the radius of gyration (Rg) of PfTubulin decreased from approximately 2.2 nm in the apo state to approximately 2.1 nm following PTX binding (Fig. 6D). This change is indicative of global structural compaction that may alter the tubulin binding interface and impair lateral protofilament interactions which is a key step in microtubule assembly (16). Such structural compaction may further undermine the formation of mitotic spindles and subpellicular microtubules during schizogony (17).

These PTX-induced structural perturbations may contribute to the observed phenotypic defects of *P. falciparum*. Schizogony arrest depends on coordinated microtubule dynamics including spindle formation for nuclear division and subpellicular microtubules for cell shape remodeling. PTX-stabilized tubulin with low RMSD values and enhanced M-loop flexibility may disrupt spindle pole organization and lead to failed nuclear segregation. This molecular mechanism is consistent with experimental observations of PTX-treated schizonts showing abnormal nuclear lobing and delayed merozoite release (Fig. 2B-E) and aligns with phenotypic changes induced by other microtubule stabilizers. For reduced merozoite formation a process that requires microtubule-mediated trafficking of apical complex components such as rhoptries and micronemes to the cell periphery compacted PfTubulin with a lower Rg value may impair the binding of kinesin motors including PfKinesin-8 and thereby block organelle transport (18). Experimentally PTX treatment reduced merozoite yield by approximately 32% from 19 to 13 per schizont and increased the proportion of unruptured schizonts by 4-fold (Fig. 2C, G). This observation is in line with findings in *Toxoplasma gondii* where PTX-induced microtubule compaction diminishes microneme secretion and host cell invasion (19).

In conclusion MD simulations support a plausible mechanistic link between PTX's molecular interaction with PfTubulin and its resultant phenotypic effects on *P. falciparum*. Specifically stabilized PTX-PfTubulin binding enhanced M-loop flexibility and global structural compaction may act in concert to disrupt the coordinated microtubule dynamics essential for parasite development. These perturbations are proposed to drive schizogony arrest and reduced merozoite

formation thereby providing support for the core antimalarial mechanism of PTX. The data further identify the PfTubulin M-loop as a promising molecular target for the rational design and optimization of next-generation antimalarial PTX derivatives...”

14. Line 713. At the start of the sentence you state "particularly pronounced synergism" and then further in the sentence it states "mild to moderate synergistic effects". It can't be both.

Reply: This contradictory description has been corrected in the revised manuscript (**Lines 954-957**).

Line 954-957: “...The DHA-PTX combination exhibited synergistic activity at sub-IC₅₀ concentrations, with CI values ranging from 0.802 to 0.954, which was indicative of mild to moderate synergism against the parasite strains...”

15. Line 717 and line 718. Significant enhancement? Again, was there a statistical test?

Reply: Corrected. We have rectified the relevant description in the revised manuscript and revised the sentence to align with the quantitative data derived from our combination index assays (**Lines 961-964**). Based on the calculated CI values, the combination of PTX and DHA exerted the most potent synergistic effect at concentrations ranging from 0.25 to 2× IC₅₀ (CI<1). Meanwhile, the combination of PTX and CQ also showed synergistic activity across the 0.5 to 1× IC₅₀ concentration range (CI<1).

Line 961-964: “...Collectively, these findings highlight a synergistic antimalarial activity in combinatorial therapeutic efficacy, positioning PTX as a promising partner drug for overcoming emerging drug resistance in malaria parasites...”

16. Line 752. Significant increase? Do you mean decrease in survival (or increase in activity)?

Reply: We are grateful to the reviewer for pointing out the ambiguity in this statement. In fact, what we intended to convey is that the combined use of PTX and DHA in the RSA can achieve a better efficacy against artemisinin-resistant parasite lines. Therefore, the survival rate obtained in the DHA monotherapy group should be higher than that in the dual-treatment group. To make this point clearer, we have revised the description as follows (**Lines 997-1002**).

Line 997-1002: "...These survival rates in DHA monotherapy were markedly higher than those observed in the PTX-DHA combination treatment group, and this elevated parasite survival indicated reduced antiparasmodial efficacy against these artemisinin-resistant strains. This notable difference further underscored the enhanced antiparasmodial activity afforded by the combinatorial regimen against such resistant parasites....."

17. Line 806 and Figure 8. Without further detail, the reader cannot trust any of the P. berghei data. In the methods section it is claimed that wildtype ANKA strain (Figs 8A+B) and a "in-house adapted resistant strain" (Figs *C+D) were used. There is no mention of what resistance the adapted parasite has in the methods (assumed to be Artemisinin-resistant from details later on). There is no mention of how resistance was generated and no data showing its resistance profile. If you look at the curves in Figs 8B+D, I see no difference in how the wildtype and resistant strains respond to DHA treatment.

Reply: Since resistance-induction assays require an extended experimental duration, typically involving 8-12 weeks of sequential drug selection, we initiated the resistance-induction experiment prior to the evaluation of PTX's antimalarial activity to accommodate this lengthy timeline. This experimental design ensured that the resistant parasite lines were comprehensively characterized before their utilization in subsequent assays.

To address this concern comprehensively, we have added the complete experimental protocol for resistance induction (including drug selection concentrations, passage frequency, and confirmation criteria) and validation data in the Supplementary information. This addition ensures full transparency of the

resistant line generation process and aligns with standard reporting practices for antimalarial resistance studies. A detailed description of this revision is provided in both the response to Reviewer 1 at **Page 4** of this file and Supplementary information.

18. Finally, the authors have not discussed at all the fact that anticancer drugs are generally quite cytotoxic - would the safety profile of PTX ever justify its use as an antimalarial?

Reply: When it comes to the well-documented cytotoxicity of PTX, we have added detailed discussions revised manuscript to elaborate on this critical issue. We have acknowledged this limitation explicitly, and further proposed the rational potential application of PTX in antimalarial therapy with specific considerations for safety and efficacy optimization (**Lines 1188-1199, Lines 1205-1209**).

Lines 1190-1201: "...Although PTX exhibited promising antimalarial potential in our experimental findings, it should also be worth noting that its inherent cytotoxicity should never be ignored. A validated strategy to mitigate such cytotoxicity is the encapsulation of PTX into biocompatible nanocarriers (e.g., albumin nanoparticles), which can effectively improve targeted delivery and reduce off-target toxic effects on normal somatic cells. In this study, we primarily aimed to explore PTX's potential as a targeted antimalarial agent; meanwhile, in consideration of its unavoidable cytotoxicity, we further sought to verify whether it could act as a synergistic partner drug in combination with ART. Therefore, we placed greater emphasis on the co-administration of PTX and DHA in both *in vitro* and *in vivo* assays, and systematically analyzed their combined antimalarial efficacy and safety characteristics..."

Lines 1205-1209: "...In addition, structural modification of PTX has also been carried out in our research, and we have successfully synthesized several PTX derivatives based on its core molecular structure. Further in-depth efforts will be made to comprehensively evaluate the antimalarial activity, cytotoxicity and structure-activity relationship of these derivatives, so as to develop safer and more effective antimalarial candidates derived from PTX..."

Re: Spectrum01957-25R1 (**Paclitaxel-Induced Tubulin Dysfunction Stalls Parasite Development: Synergistic Potential with Artemisinin against Resistant Strains**)

Dear Dr. Xinyu Yu:

Your manuscript has been accepted, and I am forwarding it to the ASM production staff for publication. Your paper will first be checked to make sure all elements meet the technical requirements. ASM staff will contact you if anything needs to be revised before copyediting and production can begin. Otherwise, you will be notified when your proofs are ready to be viewed.

Sincerely,
Jian Li
Editor
Microbiology Spectrum